# Organic laser power converter for efficient wireless micro power transfer

Yafei Wang[1,2], Zhong Zheng[1,3] ✉, Jianqiu Wang[1], Pengqing Bi[1], Zhihao Chen[1], Junzhen Ren[1,2], Cunbin An[1], Shaoqing Zhang [1,3] & Jianhui Hou [1,2,3] ✉

Wireless power transfer with collimated power transmission and efficient conversion provides an alternative charging mode for off-grid and portable micro-power electronics. However, charging micro-power electronics with low photon flux can be challenging for current laser power converters. Here we show laser power converters with organic photovoltaic cells with good performance for application in laser wireless power transfer. The laser selection strategy is established and the upper limit of efficiency is proposed. The organic laser power converters exhibit a 36.2% efficiency at a 660 nm laser with a photon flux of $9.5\,mW\,cm^{-2}$ and achieve wireless micro power transfer with an output of 0.5 W on a 2 meter scale. This work shows the good performance of organic photovoltaic cells in constructing organic laser power converters and provides a potential solution for the wireless power transfer of micro-power electronics.

Wireless power transfer (WPT) technology provide a powerful solution for efficient wireless charging, and has emerged as the focal point in the energy field[1–5]. Over long-term development, achieving high power conversion efficiency (PCE) under intense photon flux has become one of the top concerns in traditional WPT technologies[1–4]. This is because it is necessary to address the charging and endurance issues of various aerospace applications such as unmanned aerial vehicles, space vehicles and robots working under extreme conditions. With the emergence of Internet of Things (IoT), however, an increasing number of electronics require an off-grid power supply[6,7], which put forward new requirements for WPT. The power supply for such electronics greatly relies on laser WPT technology[8,9] which can guarantee high PCE over meters. Owing to the ability of collimated power transmission[1–5], the laser-based WPT is a promising power-supplying technology for micro-power electronics if the laser power converter (LPC) can exhibit broad laser wavelength applicability and high PCE under low photon flux. The target application scenario is divided by the order of WPT photon flux and varies greatly: $10^3$ to $10^6\,mW\,cm^{-2}$ for WPT techniques that often participate in military and spatial applications[4,10–16]; $10^2$ to $10^3\,mW\,cm^{-2}$ for WPT techniques are suitable for powering high-energy-consuming loads such as cameras or small unmanned aerial vehicles; and $10^{-1}$ to

$10^2\,mW\,cm^{-2}$ for a few products that have appeared in recent years due to the low-energy-consuming flexible electronics that have entered rapid development period (shown in Supplementary Note 1). Some micro sensors, detectors and chips only require a few milliwatts or even microwatts of power per use, and they do not potentially require a long-term power supply, such as anti-theft or anti-counterfeiting keys, bank cards (or other important cards), passive electronic tags (the ink screen does not consume power during daily display) and onboard ETC etc. Except for these consumer electronics, the energy consuming of long-term underwater detectors are usually as low as milliwatt-scale. The charge for these electronics could be convenient when using power beaming. Therefore, the importance of developing LPC suitable for $10^{-1}$ to $10^2\,mW\,cm^{-2}$ WPT will unfold rapidly as the applications of IoT expand.

The attainment of WPT at the target photon-flux-range ($10^{-1}$ to $10^2\,mW\,cm^{-2}$) requires developing new LPCs. Current LPCs are usually composed of silicon or gallium arsenide photovoltaic cells, which are good at obtaining high PCE at ultrahigh photon flux[2–4,17]. As photon flux decreases, the PCE of silicon or gallium arsenide photovoltaic cells drops due to the entropic loss induced intrinsic losses[2,3,18]. Moreover, the laser wavelength for most current LPCs is located in the near-

[1]State Key Laboratory of Polymer Physics and Chemistry, Beijing National Laboratory for Molecular Sciences, Institute of Chemistry, Chinese Academy of Sciences, Beijing 100190, China. [2]University of Chinese Academy of Sciences, Beijing 100049, China. [3]School of Chemistry and Biology Engineering, University of Science and Technology Beijing, Beijing 100083, China. ✉e-mail: zhongzheng@ustb.edu.cn; hjhzlz@iccas.ac.cn

infrared (NIR)[2–4,17–20], which limits their applications in the visible region. In addition, the high cost of silicon or gallium arsenide seriously reduces the energy budget of WPT for micro-power applications. Therefore, it is of great importance to develop the LPCs based on new photovoltaic technology with low cost, broad laser wavelength applicability and high PCE under micro-power density. Therefore, organic photovoltaic (OPV) cells act as a promising candidate for LPCs[21,22]. OPV is an environmentally friendly photovoltaic technology with non-toxic, lightweight, flexible, semi-transparent, solution processing, and low cost[23]. The bulk heterojunction (BHJ) layer, which serves as the charge-generation component in OPVs, exhibits the superiority in achieving high PCE under weak illumination, which is suitable for WPT with low photon flux[24,25]. Moreover, the versatile absorption spectra of conjugated polymers provides OPVs with a much broader laser selection[26,27]. Recently, an underwater OPV has been reported, and the performance is stable. However, the current OPV cells are developed for utilizing solar radiation, disregarding the construction of LPCs.

Here we report an OPV-based LPCs (OLPCs) with a typical BHJ of PBDB-TF:BTP-eC9 and illustrate the optical characteristics under lasers with various wavelengths and power densities. We find that the wavelength and power density of the laser signification dominate the photovoltaic performance of the OLPCs. In terms of exciton behaviors, the diffusion coefficient in BTP-eC9 is much higher than that in PBDB-TF; the hole transfer between BTP-eC9 and PBDB-TF is more efficient than the electron transfer; and the dissociation of excitons generated by lower energy photons produces low energy loss and restrained carrier recombination. Therefore, the use of a laser with a photon energy closer to the energy level of the charge transfer (CT) state is helpful for achieving high PCE, which is confirmed by the 33.9% PCE of PBDB-TF:BTP-eC9-based OLPC at 809 nm with 14.5 mW cm$^{-2}$. Based on the above results, we analyze the theoretical limit for the PCE of OLPC and propose a strategy for improving the PCE. Furthermore, we substituted PBDB-TF:BTP-eC9 with a wide band gap BHJ (PB2:GS-ISO) and a 36.2% PCE is achieved at 660 nm with a 9.5 mW cm$^{-2}$ photon flux. Under this illumination, the OLPC exhibits good stability at 25 °C. Finally, a 0.5 W, 2 m WPT is achieved on a 20 cm$^2$ PB2:GS-ISO-based OLPC. Our study fills the gap in the application of organic photovoltaic cells in WPT and manifests the practical potential of OLPC for low photon flux WPT.

## Results and discussion

### Basic description of the PBDB-TF:BTP-eC9 based OLPC in WPT

The WPT system with OLPC is shown in Fig. 1a. The PBDB-TF and BTP-eC9 with the illustrated structures act as the electron donor and acceptor respectively, both compose the BHJ with nanoscale phase separation. The fabrication method together with the representative current density ($J$)-voltage ($V$) curves of the OPV cells under illumination of air mass 1.5 global (AM 1.5 G), 100 mW cm$^{-2}$ are provided in Methods and Supplementary Fig. 3a. Due to the higher PCE in WPT (Table 1 and Supplementary Fig. 4), the thickness of the BHJ in the OLPC is fixed at 100 nm in this study. Since the absorption spectra of PBDB-TF and BTP-eC9 are highly complementary (Fig. 1b), the use of lasers with different wavelengths, for example, $\lambda_{533}$ and $\lambda_{809}$ would primarily excite PBDB-TF or BTP-eC9 in BHJ. Although the external quantum efficiencies (EQEs) at 533 nm and 809 nm are very similar, the photovoltaic performances are different. Figure 1c–f shows the laser intensity ($I_0$)-dependent open circuit voltage ($V_{oc}$), short circuit current density ($J_{sc}$), fill factor (FF) and PCE (the corresponding $J$-$V$ curves are provided in Supplementary Fig. 3). All investigated parameters of OLPCs under $\lambda_{809}$ illumination are higher. On the other hand, with increasing $I_0$, the $V_{oc}$ and $J_{sc}$ of OLPCs linearly increase, while the FFs and PCEs reach maximums values. PCEs of 19.4% and 33.9% are achieved in 30.2 mW cm$^{-2}$ at $\lambda_{533}$ and 14.5 mW cm$^{-2}$ at $\lambda_{809}$,

respectively, thus, more efficient photovoltaic processes occur when using $\lambda_{809}$ in the studied $I_0$ range.

### The principle of laser wavelength selection for a given OLPC

At the exciton generation stage, the excessive vibrational relaxation of excitons from the highly excited state to the ground state (GS)[28] produces inevitable energy loss when the photon energy is larger than the band gap ($E_g$) of the BHJ. Because the photon energy of $\lambda_{533}$ (2.3 eV) is 52.3% higher than that of $\lambda_{809}$ (1.5 eV), less energy loss in excitonic vibrational relaxation is likely the main reason for the higher PCE at $\lambda_{809}$. However, there is an obvious difference (3%) between the $J_{sc}$ ratio (1.6) and the photon flux ratio (1.5), when using identical $I_0$ at $\lambda_{533}$ and $\lambda_{809}$. Therefore, other intrinsic differences exist in the charge generation and recombination processes in the BHJ illuminated by $\lambda_{533}$ and $\lambda_{809}$.

The differences originate from exciton annihilation and carrier recombination. The exciton dissociation into free charge carriers mainly occurs at the donor/acceptor interface[29]. As the quantity related to exciton diffusion, the exciton diffusion length ($L_D$)[22] in $\lambda_{533}$-excited PBDB-TF (Fig. 2a) and $\lambda_{809}$-excited BTP-eC9 (Fig. 2b) can be accurately calculated (Supplementary Note 5). The $L_D$ of PBDB-TF and BTP-eC9 are 27 and 31 nm, respectively. A longer $L_D$ of BTP-eC9 correlates to more efficient exciton diffusion and illustrates the superiority of $\lambda_{809}$[30].

When illuminated by $\lambda_{533}$, a part of the exciton in the highly excited state can dissociate directly into free carriers through a "hot process" without experiencing vibrational relaxation[31]. Through the hot process, the electron in the exciton can quickly transfer to the CT manifold even at low temperature ($T$), which can be clearly observed in the higher $V_{oc}$ of the OLPC under $\lambda_{533}$ (Fig. 2c and Supplementary Fig. 8)[31]. In the BHJ of PBDB-TF:BTP-eC9, however, the electron dissociated through the hot process more easily decays to GS, which leads to geminate recombination. The minimal activation energy ($E_a$) to overcome geminate recombination can be determined by the slope of $ln$(EQE) to vs. $1/k_BT$ in EQE/$T$ curves (Fig. 2d and Supplementary Note 6)[32], where the $k_B$ is Boltzmann constant. For the BHJ illuminated by $\lambda_{533}$ and $\lambda_{809}$, the $E_a$ values are 0.5 and 0.3 meV in the range of 77–110 K, and becomes 1.5 and 0.8 meV in the range of 110–300 K, respectively. Therefore, stronger geminate recombination occurs when illuminated by $\lambda_{533}$. Furthermore, stronger bimolecular recombination in OLPC at $\lambda_{533}$ can also be resolved from the larger slope of the $V_{oc}$-$I_0$ dependence in Fig. 1c. A lower slope of the OLPC at $\lambda_{809}$ illustrates the efficient carrier migration.

Figure 3a shows the schematic illustration of the hole and electron transfer induced by the dissociation of excitons generated in PBDB-TF and BTP-eC9. In terms of the donor/acceptor interface, since the highest occupied molecular orbital (HOMO) of BTP-eC9 is deeper than that of PBDB-TF, exciton dissociation is accompanied by hole transfer, which can be traced by the lifetime of the excitons at the donor/acceptor interface[33]. Excitation wavelength of 533 and 809 nm are used to selectively excite PBDB-TF and BTP-eC9 in the BHJ respectively, and the two-dimensional TA spectra are shown in Fig. 3b,c. Representative TA spectra at the indicated delay times are shown in Fig. 3d, f. The signals at 630 and 810 nm originate from ground-state bleaching (GSB) of the optically excited exciton transition in PBDB-TF and BTP-eC9[33]. The excited-state absorptions (ESA) appear at 675 nm and 890 nm are attributable to PBDB-TF and BTP-eC9. The ESA, appears at 1400–1600 nm, is considered to the intermediate intra-moiety excited (i-EX) state of the hole transfer channel[33]. Upon excitation at 533 and 809 nm, the $\Delta A$ at 630 and 810 nm are obtained, which show the decay kinetics of PBDB-TF and BTP-eC9, respectively (Fig. 3e, g). By fitting the decay kinetics with biexponential modeling (Supplementary Table. 3), the time constants ($\tau_1$ and $\tau_2$) are obtained, and can be assigned to the time scales of ultrafast exciton dissociation and exciton diffusion lifetime, respectively[33]. For excitation at 533 nm, $\tau_1$ and $\tau_2$ are 1.0 and

**Table. 1 | The photovoltaic parameters of OLPCs based on PBDB-TF:BTP-eC9 with various BHJ thicknesses**

| Light sources | $I_0$ (mW cm$^{-2}$) | BHJ thickness (nm) | $V_{oc}$ (V) | $J_{sc}$ (mA cm$^{-2}$) | FF (%) | PCE (%) |
|---|---|---|---|---|---|---|
| AM 1.5 G | 100.0 | 70 | 0.848 | 25.9 | 77.8 | 17.1 |
| | | 100 | 0.845 | 26.6 | 77.4 | 17.4 |
| | | 150 | 0.843 | 27.0 | 75.2 | 17.1 |
| $\lambda_{533}$ | 10.7 | 70 | 0.752 | 3.2 | 76.5 | 1.8 |
| | | 100 | 0.749 | 3.4 | 75.8 | 1.9 |
| | | 150 | 0.746 | 3.5 | 72.7 | 1.9 |
| | 63.6 | 70 | 0.805 | 19.2 | 76.5 | 11.8 |
| | | 100 | 0.802 | 19.8 | 76.0 | 12.1 |
| | | 150 | 0.798 | 19.9 | 73.0 | 11.6 |
| | 125.6 | 70 | 0.827 | 36.5 | 75.0 | 22.6 |
| | | 100 | 0.824 | 37.3 | 74.6 | 22.9 |
| | | 150 | 0.822 | 37.5 | 71.9 | 22.2 |
| $\lambda_{809}$ | 14.5 | 70 | 0.820 | 7.4 | 80.1 | 4.9 |
| | | 100 | 0.817 | 7.6 | 79.5 | 4.9 |
| | | 150 | 0.815 | 7.6 | 77.2 | 4.8 |
| | 47.0 | 70 | 0.845 | 21.2 | 79.0 | 14.2 |
| | | 100 | 0.843 | 21.8 | 78.4 | 14.4 |
| | | 150 | 0.840 | 22.0 | 74.8 | 13.8 |
| | 99.9 | 70 | 0.864 | 43.0 | 77.1 | 28.6 |
| | | 100 | 0.861 | 44.1 | 76.9 | 29.2 |
| | | 150 | 0.857 | 44.6 | 72.9 | 27.9 |

24.1 ps, respectively; For excitation at 809 nm, $\tau_1$ and $\tau_2$ are 1.7 and 26.5 ps, respectively. The time scales of $\tau_1$ and $\tau_2$ in this case are both comparable to the values of effectively performing BHJs reported elsewhere[33–36]. According to results from the literature[37,38], the efficient exciton self-excitation dissociation in non-fullerene acceptor molecules facilitates the generation of free carriers and therefore the use of $\lambda_{809}$ in WPT is helpful for improving the PCE. In conjugated systems, the hole transfer can be highly efficient[33,39], which is observed in our study with the similar $\tau_1$ in Fig. 3e, g[25]. The efficient hole transfer enables the application of $\lambda_{809}$ in the WPT.

Based on the above discussions regarding $L_D$, $E_a$, $\tau_1$ and $\tau_2$, we establish a laser wavelength selection principle for the OLPCs. The photon energy needs to be higher than the $E_g$ of the BHJ and the corresponding wavelength needs to be located within the high EQE region. Beyond these two preconditions, the difference between the photon energy of the laser and the $E_g$ of the BHJ needs to be as small as possible, such as $\lambda_{809}$ for PBDB-TF:BTP-eC9. This deduction can be verified by the highest PCEs when using different laser wavelengths (Supplementary Table. 1).

## PCE prediction for the OLPCs

Therefore, we select another BHJ, PB2:GS-ISO (Fig. 4c), to fabricate the OLPC. As shown in Fig. 4d, the absorption onset locates at ~700 nm, indicating a 1.8 eV $E_g$. The EQE of the PB2:GS-ISO-based OPV cell is over 70%, ranging from 400 to 660 nm. According to the aforementioned wavelength selection principle, $\lambda_{660}$ should be suitable for this OLPC. As shown in Fig. 4e and Table 2 (The J-V curves are shown in Supplementary Fig. 11), ranging from 1.9 to 211.2 mW cm$^{-2}$ at $\lambda_{660}$, the PB2:GS-ISO-based OLPC exhibits over 36% PCEs, with a maximum of 36.2% at 9.5 mW cm$^{-2}$. These results confirm that the use of a BHJ with a large $E_g$

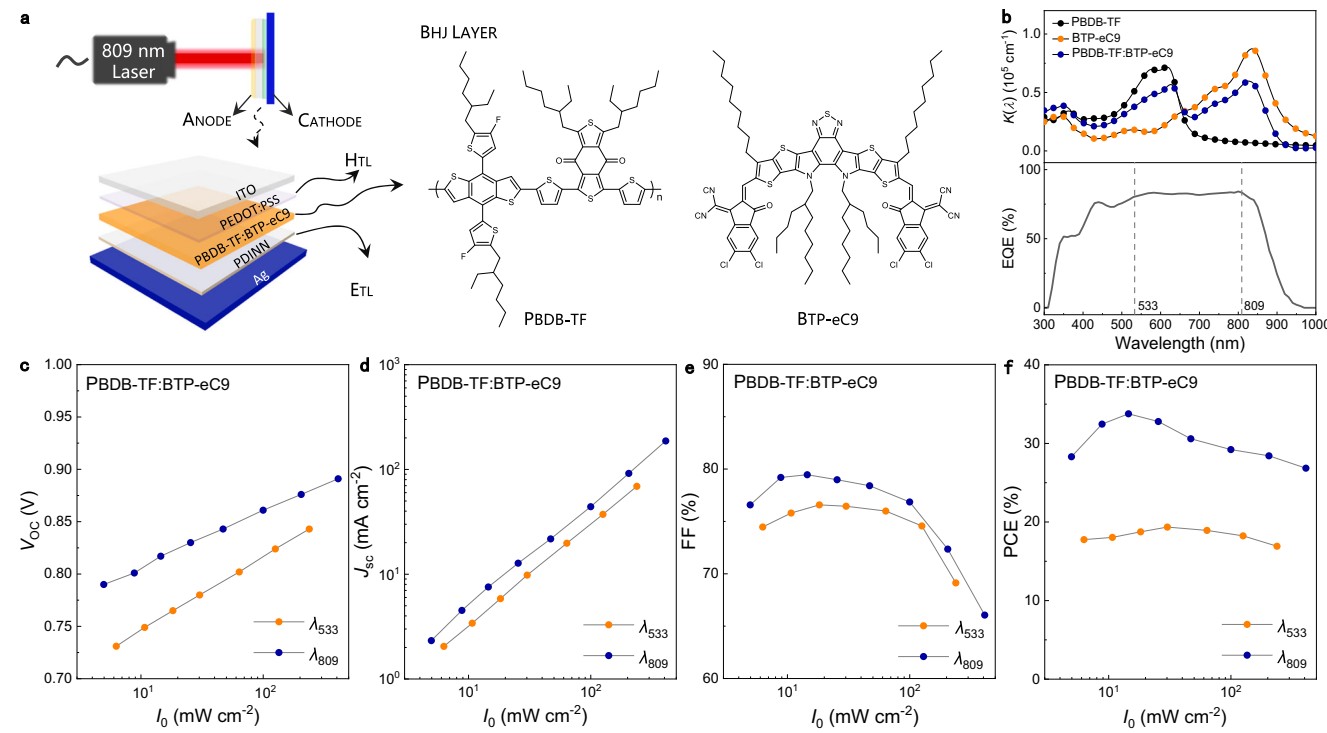

**Fig. 1 | Basic description of the OLPC in WPT. a** Diagrams of the WPT devices. Devices with a sandwich-structure similar to the OPV cells for AM 1.5 G 100 mW cm$^{-2}$ are used for the OLPCs. The middle part illustrates the molecular structures of PBDB-TF and BTP-eC9. The indium tin oxide, hole transporting layer and electron transporting layer are labeled with ITO, HTL and ETL, respectively. **b** Upper item displays the absorption coefficient of neat PBDB-TF, BTP-eC9 and PBDB-TF:BTP-eC9 BHJ, respectively. The bottom item displays the EQE spectra of the OPV under AM 1.5 G illumination at 100 mW m$^{-2}$. The spectrum line shapes of the lasers used in this work are shown in Supplementary Fig. 2. $I_0$-dependent **c**, $V_{oc}$, **d**, $J_{sc}$, **e**, FF and **f**, PCE of the OLPC under lasers with different wavelengths. The lasers are labeled with $\lambda$ and their wavelengths marked in subscripts.

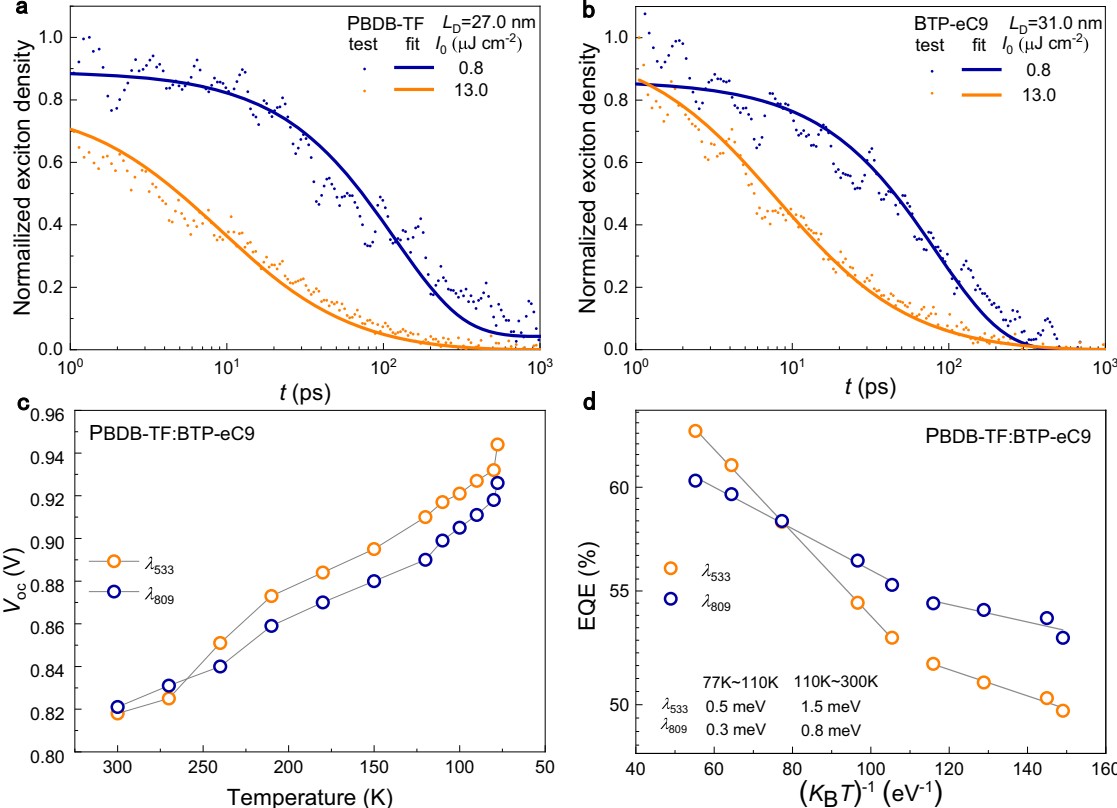

**Fig. 2 | Diffusion and dissociation of excitons generated by $\lambda_{533}$ and $\lambda_{809}$.** Singlet-singlet exciton annihilation (SSA) decay in neat films of **a**, PBDB-TF (excitation wavelength is 533 nm) and **b**, BTP-eC9 (excitation wavelength is 809 nm). Detailed information of SSA modeling is listed in Supplementary Note 5. The transient absorption (TA) 2D-images of neat films are shown in Supplementary

Fig. 5. The *T*-dependent **c**, $V_{oc}$ and **d**, EQE of the OLPC excited at 533 and 809 nm. The $I_0$ of the 533 and 809 nm excitation lights are controlled to 26.8 and 17.2 mW cm$^{-2}$, respectively, because under these conditions, both the saturated photocurrent densities ($J_{sat}$) of the OLPC are -15.2 mA cm$^{-2}$ and the influence of the carrier density on $V_{oc}$ is negligible.

is very helpful for improving the PCE of the OLPC. The $L_D$ and the processes of hole and electron transfer induced by the dissociation of exciton generated in PB2 and GS-ISO were further studied by TA (The details are attached in the SI.). According to Fig. 4f,g, the calculated $L_D$ of PB2 and GS-ISO are 11.8 and 27.6 nm, respectively. Figure 4h,i shows the TA data recorded from the blend PB2:GS-ISO, when PB2 was selectively excited at 533 nm and GS-ISO was selectively excited at 660 nm. The decay kinetics of PB2 and GS-ISO are obtained the $\Delta A$ at 630 and 810 nm (Fig. 4j), which were excited at 533 and 809 nm, respectively. The faster $\tau_1$ and $\tau_2$ for excitation at 660 nm demonstrated that the more efficient hole transfer in PB2:GS-ISO.

In practical WPT, the output power ($P_{out}$) of the OLPC is another critical factor. Figure 4e shows that under increasing laser intensity, the $P_{out}$ of the PB2:GS-ISO-based OLPC increases to 211.2 mW cm$^{-2}$ while the FF significantly decreases to 64.6%. To determine the reason for the PCE reduction when increasing $I_0$, we measured the series resistance ($R_s$) of pure electrodes (Supplementary Fig. 13). The $R_s$ of whole OLPC decreases as $I_0$ increases and approaches the $R_s$ of pure electrodes (5.6 Ω cm). Therefore, the proportion of $R_s$ induced by electrodes gradually increases as $I_0$ increases. Thus, the heat loss of electrodes acts as the major reason for the PCE decrease. A distinction can be drawn between this issue and the potential of OPV in WPT because the electrode or device configuration can potentially be addressed by further engineering.

The PB2:GS-ISO-based OLPC shows good stability under continuous illumination at $\lambda_{660}$, 9.5 mW cm$^{-2}$. With *T* controlled at 25 °C (using the holder in Supplementary Fig. 14), the extrapolated T80 lifetime (the time required to reach 80% of initial performance) is as long as 7300 h. With respect to practical applications, lasers can be

replaced by inexpensive but safe monochromatic light-emitting diodes (LED) spotlights. Coupled by a simple collimator, the WPT on 2 m scale can be achieved. As shown in Fig. 4e, the *J-V* curve of the PB2:GS-ISO-based 20 cm$^2$ OLPC displays a 26.2% PCE, and $P_{out}$ is as high as 0.5 W when using a 2-m-away $\lambda_{660}$ 100 mW cm$^{-2}$ LED. The performance of the LED charging system can fulfill the requirements of micro-power electronics, such as passive electronic tags, on-board ETCs and microfluidic chips. As shown in Supplementary Note 9, OLPC is still indispensable owing to the advantage in energy budget and laser wavelength adaptability. Based on detail investigation, we summarized the bandgap tunability, flexibility, contain heavy metals or not, power per weight and cost in Supplementary Table 6. We believe that the features of highly adjustable bandgap, ultra-high flexibility, avoidance of heavy metals, high power-per-weight, and low cost enable OLPCs in future wireless power transfer.

We demonstrate OLPCs with OPV cells for WPT technology with good performance at low optical intensities. Moreover, this work illustrates the applicability of the OPV cells as LPCs for wireless micro-power transmission. The wavelength of laser determines the absorption component and thus the excitonic properties. We propose the laser selection strategy based on systematic photophysical studies. In PBDB-TF:BTP-eC9-based OLPC with $\lambda_{809}$ 14.5 mW cm$^{-2}$, the PCE is 33.9%, which is much higher than the PCE of OPV under AM 1.5 G 100 mW cm$^{-2}$. Based on fundamental calculations, the relationships among laser wavelength, EQE, $V_{loss}$ and PCE have been determined, and a rational PCE prediction is carried out. By using a BHJ with 1.7 $E_g$ (PB2:GS-ISO), a 36.2% PCE and a 7300 h T80 are obtained in 0.04 cm$^2$ OLPC with $\lambda_{660}$ at 9.5 mW cm$^{-2}$. The FF decrease at a larger $I_0$ primarily originates from the $R_s$ of the electrodes, which could be addressed by

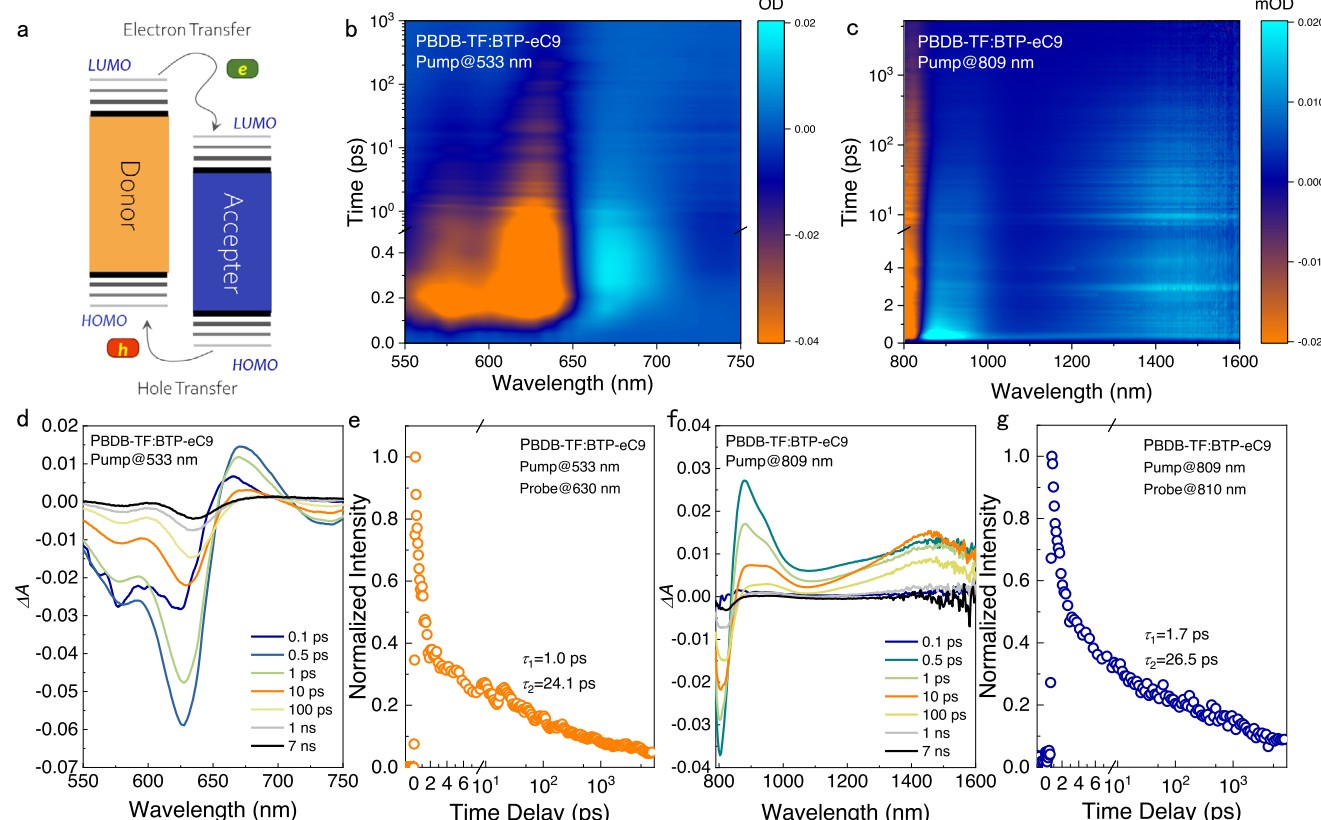

**Fig. 3 | Hole and electron transfer processes in the BHJ. a** Schematic illustration of the hole and electron transfer induced by the dissociation of excitons generated in PBDB-TF and BTP-eC9. The transient absorption (TA) two dimensional images of PBDB-TF:BTP-eC9 BHJ under excitation of (**b**). 533 nm, 10.0 μJ cm$^{-2}$ and **c**, 809 nm, 10.0 μJ cm$^{-2}$. The TA spectra at different delay times of the PBDB-TF:BTP-eC9 film with excitation at **d**, 533 nm and **f**, 809 nm, respectively. Normalized femtosecond TA exciton dynamics for the PBDB-TF:BTP-eC9 film excited at **e**, 533 nm with 10.0 μJ cm$^{-2}$ (probed at 630 nm) and **g**, 809 nm with 10.0 μJ cm$^{-2}$ (probed at 810 nm).

further collector design. By using an inexpensive but safe collimated LED, a WPT with 0.5 W $P_{out}$ is realized by PB2:GS-ISO-based 20 cm$^2$ OLPC in 2 meter under $\lambda_{660}$ 100 mW cm$^{-2}$. Due to the adjustable bandgap, flexibility, absence of heavy metals, high power per weight, and low cost, OLPCs have unique advantages in wireless power transfer.

## Methods

### Materials
PBDB-TF, BTP-eC9, PB2, GS-ISO and PDINN were purchased from Solarmer Materials Inc and used as received. Poly(3,4-ethylenedioxythiophene):polystyrene sulfonate (PEDOT:PSS, CLEVIOS™ PVP AI 4083) was purchased from Heraeus Inc. All solvents were commercially available from Acros. Glass/indium tin oxide (ITO) (12 Ω Υ$^{-1}$) was purchased from South China Xiang's Science & Technical Company Limited.

Device Fabrication. PBDB-TF:BTP-eC9 (wt/wt, 1:1.2) was dissolved in CF at a concentration of 8 mg ml$^{-1}$ with respect to the polymer. The solution was stirred at 40 °C for at least 2 h. Then a small amount of 1,8-diiodooctane (DIO) (v/v, DIO/CF) was added 30 min prior to the spin coating process. ITO substrates were sequentially cleaned with detergent, deionized (DI) water, acetone and isopropanol by sonication. The dried ITO substrates were treated with UV-ozone for 15 min, and then the PEDOT:PSS precursor was spin-coated on the surface of the ITO. The rotation speed needed to be tuned to obtain a 30 nm thick film, followed by 160 °C annealing in air. Subsequently, the substrates with annealed PEDOT: PSS were transferred to the glove box. The PBDB-TF:BTP-eC9 solution was spin-coated on PEDOT:PSS layers, and then the films were thermally

annealed at 100 °C for 10 min. PDINN was spin-coated on the annealed active layers to ~5 nm. After, a 100 nm Ag (ZhongNuo Advanced Material (Beijing) Technology Co., Ltd.) cathode was thermally evaporated under high vacuum (ca. 1 × 10$^{-5}$ Pa). The device area, as defined by the overlap of the ITO and Al, was 0.04 cm$^2$. PB2:GS-ISO (wt/wt, 1:1.3) BHJ was dissolved in CB at a concentration of 10 mg mL$^{-1}$ with respect to the polymer. Then, a small amount of 1,8-diiodooctane (DIO) (v/v, DIO/CB) was added 30 min prior to the spin coating process. The temperature of thermal annealing of the active layer was 150 °C for 10 min. The device areas were 0.037 cm$^2$ and 2 cm$^2$. All devices were encapsulated with epoxy resin before the experimental test. Neat films of PBDB-TF and BTP-eC9 were prepared from chloroform (10 mg mL$^{-1}$ and 15 mg mL$^{-1}$) and spin coated onto spectrosil fused silica substrates under N$_2$ at room temperature at 3000 rpm. The blend films of PBDB-TF: polystyrene and BTP-eC9:polystyrene were prepared from CF solutions of PBDB-TF and BTP-eC9 (2 mg mL$^{-1}$) and polystyrene (54 mg mL$^{-1}$) under the same conditions.

Fabrication of 20 cm$^2$ OPV modules. Glass/ITO/PEDOT:PSS/ PB2:GS-ISO/PFN-Br/Al were fabricated by blade coating method. The ITO substrates were purchased from Huananxiangcheng Inc, and patterned with 12 μm P1. ITO substrates were cleaned by above method. The area of mask is 20 cm$^2$. After coating and annealing the active layer, the PFN-Br was coated. P2 pattern was formed by a mechanical scribing machine with 50 μm scribing blade. The samples were transferred to thermal evaporator and 150 nm of Ag were deposited. Then, the P3 patterns were formed by a mechanical scribing machine. Notable, to prevent the etched silver from sticking and causing a short circuit, the air knife was applied during the etching process.

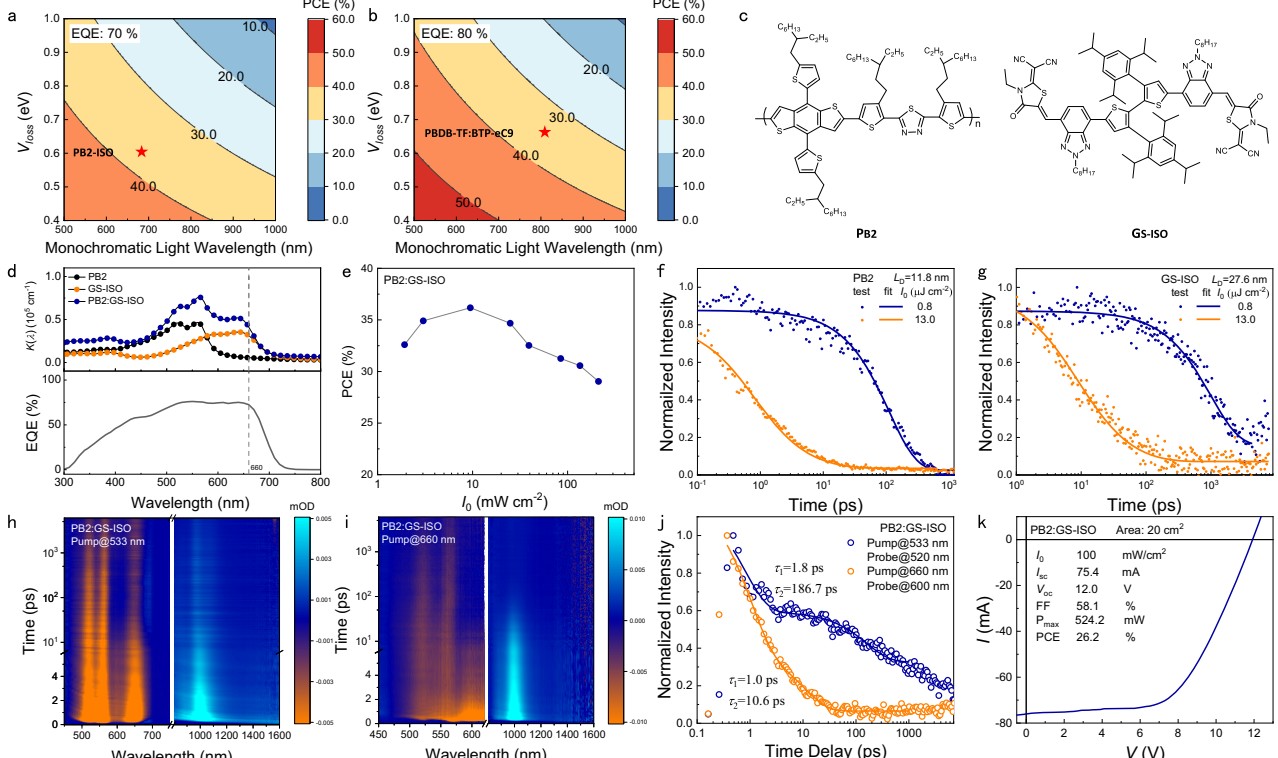

**Fig. 4 | The band gap tunability of the OLPCs and the PCEs predictions under various assumptions.** Relationship among PCE, $V_{loss}$ ($V_{loss} = 1240/\lambda \cdot qV_{oc}$, where $q$ is the electron charge) and **a**, EQE 70% and **b**, EQE 80%, assuming that the $E_g$ of the BHJ can be tuned to the value of the photon energy of the laser. (details of calculation are provided in Supplementary Note 7). The points for the studied OLPCs are marked in this figure. **c**, Molecular structures of PB2 and GS-ISO. **d**, Top figure: the absorption spectra of neat PB2, GS-ISO and the BHJ. Bottom figure: EQE of PB2:GS-ISO. **e**, The PCE PB2:GS-ISO OLPC at $\lambda_{660}$ with various $I_0$. The SSA decay in neat films of **f**, PB2 (excitation wavelength is 533 nm) and **g**, GS-ISO (excitation wavelength is 660 nm). The TA 2D-images of neat films are shown in Supplementary Fig. 6. **h** The TA two dimensional images of PB2:GS-ISO BHJ under excitation of **i** 533 nm, 10.0 μJ cm$^{-2}$ and **c** 660 nm, 10.0 μJ cm$^{-2}$. **j** Normalized femtosecond TA exciton dynamics excited at 533 nm with 10.0 μJ cm$^{-2}$ (probed at 520 nm) and 660 nm with 10.0 μJ cm$^{-2}$ (probed at 600 nm) for PB2:GS-ISO film. **k** The *J-V* curve of the 20 cm$^2$ OLPC in 0.5 W WPT at 660 nm LED, and the schematic diagram of 20 cm$^2$ module is shown in Supplementary Fig. 12.

**Table. 2 | The photovoltaic parameters of the OLPC with PB2:GS-ISO BHJ under different light sources**

| Light source | $I_0$ (mW cm$^{-2}$) | $V_{oc}$ (V) | $J_{sc}$ (mA cm$^{-2}$) | FF (%) | PCE (%) | EQE (%)[a] |
|---|---|---|---|---|---|---|
| AM 1.5 G | 100.0 | 1.230 | 12.9 | 68.2 | 10.9 | – |
| $\lambda_{660}$ | 1.9 | 1.120 | 0.7 | 79.8 | 32.6 | 68.9 |
| | 3.0 | 1.140 | 1.2 | 80.4 | 34.9 | 71.4 |
| | 9.5 | 1.170 | 3.6 | 81.3 | 36.2 | 71.4 |
| | 24.9 | 1.190 | 9.7 | 74.8 | 34.7 | 73.2 |
| | 39.1 | 1.210 | 14.6 | 72.2 | 32.5 | 70.0 |
| | 84.7 | 1.230 | 30.8 | 69.8 | 31.3 | 68.4 |
| | 134.8 | 1.250 | 49.3 | 66.9 | 30.6 | 68.7 |
| | 211.2 | 1.260 | 75.3 | 64.6 | 29.0 | 67.0 |

[a]EQE calculated by $J_{sc}$.

## *J-V* characterization

The *J-V* characteristics of solar cells were recorded with a Keithley 2400 source meter unit under standard AM 1.5 G (100 mW cm$^{-2}$) solar irradiation sourced via a solar simulator (SS-F5-3A, Enli Technology CO., Ltd.) The AM 1.5 G light source with a radiative intensity of 100 mW cm$^{-2}$ was calibrated by a standard silicon solar cell (SRC-2020), this standard cell was calibrated by the National Institute of Metrology (NIM), China. The laser generators for the *J-V* characterization were purchased from Beijing Blueprint Photoelectricity

Technology Co., Ltd. The illumination intensities of the laser sources were measured by power meter (S142C, Thorlabs). The *J-V* curves were measured from −1.5 to 1.5 V, with a scan step of 50 mV and a dwell time of 5 ms. The active area was defined by a metal mask with a 0.0223 cm$^2$ aperture. The cells were characterized at room temperature in glove box filled under a N$_2$ atmosphere.

## UV−Vis absorption measurements

The spectra of the materials were measured by a UV-Vis spectro-photometer (UH5300, Hitachi).

## EQE measurements

The EQE data were obtained from a solar cell spectral response measurement system (QE-R3011, Enlitech Technology Co. Ltd).

## Laser spectrum line shape measurements

The spectral line shapes of the lasers were measured by a spectrometer (OCEAN-FX-VIS-NIR-ES) purchased from Ocean Optics Inc.

## TA measurements

Transient absorption (TA) spectra were measured by the Ultrafast Helios pump-probe system in conjunction with a regenerative amplified laser system from Coherent. A Ti:sapphire amplifier (Astrella, Coherent) was used to generate an 800 nm pulse with a repetition rate of 1 kHz, a length of 100 fs, and an energy of 7 mJ pulse$^{-1}$. Then, the 800 nm pulse was separated into two parts by a beam splitter. One part was coupled into an optical parametric amplifier (TOPAS,

Coherent) to generate the pump pulses at various wavelengths. The other part was focused onto a sapphire plate and a YAG plate to generate white light supercontinuum as the probe beams. The spectral ranges of the probe beams were 420–800 nm and 750–1600 nm, respectively. The time delay between the pump and probe was controlled by a motorized optical delay line with a maximum delay time of 8 ns. The sample films were spin-coated onto the quartz plates and prepared to resist water and oxygen in air, the samples were encapsulated by epoxy resin in $N_2$ filled glove box. The pump pulse was chopped by a mechanical chopper at 500 Hz and then focused onto the mounted sample with probe beams. The probe beam was collimated and focused into a fiber-coupled multi-channel spectrometer with a CCD sensor. The energy of the pump pulse was measured and calibrated by a power meter (PM400, Thorlabs). To avoid exciton-exciton and exciton-charge annihilation effects, the excitation densities of the lasers used for TA measurements were kept as low as 10 µJ cm$^{-2}$.

## Stability measurement

The OLPC devices were encapsulated by exposing epoxy resin to a 365 nm UV lamp for 10 min and further tested in air (45% humidity). An array of 660 nm light-emitting diodes (LEDs) was utilized as the light source and the light intensity was 9.5 mW cm$^{-2}$. The light intensity was set to the same intensity as the output from the optimal PCE. During storage (open-circuit conditions) and measurement, the devices were kept at 25 °C. The area of aperture in front of the OLPC was 0.0223 cm$^2$.

## Reporting summary

Further information on research design is available in the Nature Portfolio Reporting Summary linked to this article.

# Data availability

The data that support the findings of this study are presented in Supplementary Information. The source data underlying Figs. 1b–f, 2a–d, 3b–g and 4a–b, d–k are provided in the Source Data files with this paper or available from the corresponding author on request. Source data are provided with this paper.

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

## Acknowledgements

J.H. would like to acknowledge the financial support from the National Natural Science Foundation of China (NSFC, 52120105005, 21835006). Z.Z. would like to acknowledge the financial support from NSFC (22275016), Beijing Municipal Science & Technology Commission (2232078) and Beijing National Laboratory for Molecular Sciences (BNLMS) Junior Fellow (2019BMS20014).

## Author contributions

Z.Z. and J.H. conceived and directed this project. Y.W. performed experiments and measurements. J.W. helped to conduct the experiments and took part in the discussion. P.B. and Z.C. carried out the TA characterizations and processed the data. S.Z., C.A., and J.R. provided photovoltaic materials of this project. This manuscript was mainly prepared by J.H., Z.Z., and Y.W., and all the authors discussed the results and commented on the manuscript.

## Competing interests

The authors declare no competing interests.
