## [Peer Review File · Nature Communications]

Organic Laser Power Converter for Efficient Wireless Micro Power TransferREVIEWER COMMENTS

Reviewer #1 (Remarks to the Author):

Wang et al. demonstrated a beamed power converter using organic solar cells under laser illumination. They employed transient spectroscopic measurements and theoretical calculations to investigate the physical mechanism underlying the power converter. This work expands the potential applications of organic solar cells and is of significant interest to the community of organic optoelectronics. However, before publication, the following issues should be addressed:

1) While the authors mention the wireless power transfer with low photon flux as a merit of the device performance, it would be helpful if they compared the performance with other available techniques.

2) On page 1, line 35-36, the authors state that "As photon flux decreases, the PCE of silicon or gallium arsenide photovoltaic cells drops due to the Auger process induced intrinsic losses". The authors should explain why this happens. In principle, the Auger process is a many-body effect that should be more significant under higher photon flux excitation. Additionally, the authors should clarify if there is any particular reason why the Auger process is different in organic devices.

3) On line 107-108, the authors mention that "Only the exciton diffuses to the donor/acceptor interface can be dissociated into free carrier," which is true in conventional devices. However, in OPV devices based on non-fullerene acceptors, charge separation is ready in individual acceptor domains, as evidenced by recent publications such as Nat. Commun. 2022, 13, 2827(2022) or Ref. 26. Therefore, the authors should consider this point.

4) The authors performed transient absorption measurements with excitation fluence of ~ 10 $\mu\text{J}/\text{cm}^2$, which is far beyond the working condition of the device. In this regime, exciton annihilation cannot be avoided. The authors should rationalize this disparity. Additionally, the authors note that the probe spectrum covers 750-1600 nm, but the data is only plotted in the range < 1100 nm. However, some essential features appear in the longer-wavelength range, and the authors should carefully check for them.

Reviewer #2 (Remarks to the Author):

The authors obtained interesting results with organic power converters for low input power conditions at different wavelengths. Unfortunately, I cannot recommend the paper for publication and significant revision would be required. Furthermore, I do not believe the results are noteworthy or significant enough to warrant publication in Nature Communications.

The grammar and the style needs significant revisions. For example, the authors use “beamed power converter” => I think they should remove “beamed”. Instead of “beamed”, maybe use “laser” or “optical”, as it is usual used in the field. Grammatically, I think “beamed power converter” does not make sense.

The authors used “highly collimated power transmission” => why “highly”, for example further towards the end of the manuscript, the authors proposed “With respect to practical application, the laser can be alternated by cheap but safe monochromatic light-emitting diode (LED).”, so the collimation certainly need not to be “high”.

The authors further claim, “current beamed power converters are incapable of charging micro-power electronics with low photon flux” => I believe this statement to be not true nor substantiated.

They then use the sentence “The principle of laser selecting is established and the up limit of efficiency is prospected.” => I find this sentence has no meaning or its meaning is very unclear.

In addition, the authors mentioned “36.18% efficiency under 660 nm laser with photon flux of 9.46 mW/cm²” => there are too many digits of precision, all throughout the manuscript, including in Table 1, etc. For example, the real efficiency accuracy is probably at best 1 decimal digit of precision. For example, “0.52 W in 2 meters scale” should be replace with “0.5 W for a distance of 2 m”.

The authors claim, “Our work for the first time manifests the superiority of organic photovoltaic cell in constructing organic beamed power converters and provides a solution for the wireless power transfer of micro-power electronics” I find this claim is misleading and unsubstantiated, for example, the literature shows that it is not superior. For example, Komuro et al published “A 43.0% efficient GaInP photonic power converter under high-power 638 nm laser irradiation of 17 W cm⁻²,” or Fafard et al have shown “At 100 mW of input power, an efficiency of Eff = 59.7% is obtained ... The output power (P_{mpp}) reaches 242 mW at 500 mW of input power and 415 mW at 1 W ”.

Furthermore, the authors probably have not demonstrated a real “solution”, because the reliability is still ambiguous. Some stability results are shown in the supplementary material, but a real solution would require a solid reliability study. In addition, the device area considered in this study is order of magnitudes much greater than the area used of the typical semiconductor power converters, for example 20 cm² vs < ~0.2cm², therefore the arguments and the comparisons must be adjusted accordingly.

The authors mentioned, “As photon flux decreases, the PCE of silicon or gallium arsenide photovoltaic cells drops due to the Auger process induced intrinsic losses”. I believe this statement is not true nor

substantiated: at low photon flux, good state-of-the-art power converters still have good PCE, and Auger process are not expected to be significant at low densities of charged carriers. The output voltage of state-of-the-art semiconductor power converters is only slightly lower at lower at low photon fluxes, as expected from the slightly lower output voltage expected from an ideal diode behavior.

The authors mentioned, “it is of great importance to develop the BPC based on new photovoltaic technology with low cost, broad laser wavelength applicability and high PCE under micro-power density.” I find the only aspect of this statement that could be valid in the context of the results presented in the paper is the “low cost” aspect. Although, no data as such to substantiate the relative low cost aspect is presented in the manuscript.

It would probably be interesting if the authors elaborated more on the stability of the results, for example, in Fig. 1, the FF decreases at optical intensities as low as a few tens of milliwatts per cm^2 . Can the authors comment regarding if the devices show degradation at these intensities? Can the I-V or FF curves be retraced multiple times over extended periods of exposure times.

Why use the scientific notation for the vertical scale of Fig. 2d?

Fig. 2 and 3 show the results of time-resolved and low temperature photocarrier experiments, while the results can be interesting to better understand the carrier dynamics in these materials, they do not demonstrate the “superiority” of the power converters or the efficient power transfer. Fig. 4e shows an I-V curve of a 20cm^2 module at 2m with an efficiency of 26%, the author should clearly mention the input wavelength for Fig. 4e, as well as the interconnection architecture that leads to a V_{oc} of $\sim 12\text{V}$. I do not see any superiority of the results presented in Fig 4e compared to other state-of-the-art power converters, other of course than potentially the low cost of the organic materials used.

Reviewer #3 (Remarks to the Author):

Re: Organic Beamed Power Converter for Efficient Wireless Micro Power Transfer, by Yafei Wang, et al.

Since the first days of organic photovoltaics (OPVs), our community has searched for applications that might demonstrate their unique advantages. Typically, OPV arguments have been based on proposed cost benefits. But over the years, some have also noted that OPVs can outperform conventional cells at low brightness.

The physical foundation for interest in performance at low brightness is the localization of optical and electrical excited states in organic materials. This can provide protection from traps, defects and recombination losses. But localization also creates problems at higher optical densities, like lower charge carrier mobilities, and losses such as exciton-exciton and exciton-charge annihilation.

To the best of my knowledge, arguments for the use of OPVs at low brightness never caught on. The question is why would anyone care about a solar cell at low brightness? Maybe there is some benefit in the morning or evening, or maybe the objective is energy harvesting from ambient light? But on the other hand (at least for OPVs), we can't ignore performance at high optical flux either because those are the crucial conditions for actually generating power.

This manuscript presents a variation on this old argument, but applied now to wireless power transfer. The OPVs are indeed outstanding: 17% under AM1.5, and even better under monochromatic excitation closer to the absorption edge. Two main materials are employed, with the wider gap choice unsurprisingly outperforming the conventional OPV structure. The OPV material system is also characterized using TA, which is nice although not especially relevant to the PB2:GS-ISO system that is highlighted as the optimal solution at 660nm.

My main concern with the manuscript is the engineering argument, especially the justification for the key conclusion 'We demonstrate the superiority of constructing OBPC with OPV cell for WPT technology.' I couldn't find any comparisons between the OPV systems and conventional cells, and while the performance sounds great in isolation, one has to wonder what GaAs can do at 660nm. Are we going to exclude the use of Si or GaAs based on cost? I wonder because WPT may not need a large area cell. I also don't understand whether low power, visible spectrum WPT is demanded by any practical application. The last line of the preceding paragraph seems crucial 'The performance of the LED charging system can fulfill the requirement of micro-power electronics such as passive electronic tag, on-board ETC and microfluid chip.' It struck me that a different way to present this work might have been to start with that target application, justify its significance with citations, explain why it needs to be visible spectrum, LED-powered, etc... and then work through the quantitative benefits of this OPV system versus conventional alternatives. I would really like to believe that OPV has a compelling application in WPT, but this manuscript unfortunately does not make that case.

Response to decision letter (purple for comments; black for responses; red for revisions).

Reviewer #1 (Remarks to the Author):

Wang et al. demonstrated a beamed power converter using organic solar cells under laser illumination. They employed transient spectroscopic measurements and theoretical calculations to investigate the physical mechanism underlying the power converter. This work expands the potential applications of organic solar cells and is of significant interest to the community of organic optoelectronics. However, before publication, the following issues should be addressed:

1) While the authors mention the wireless power transfer with low photon flux as a merit of the device performance, it would be helpful if they compared the performance with other available techniques.

Response: We appreciate the comment. We compared the performance in low photon flux WPT by using single-junction gallium arsenide (GaAs), perovskite (PVK) and monocrystalline silicon (Si) photoelectric converter. The results are listed here (reput in supporting information **Section S9**).

Fig. R1. The EQE spectra of (a) Gallium arsenide (GaAs), (b) perovskite (PVK) and (c) silicon (Si) under the illumination of AM 1.5G 100 mW/m². The *J*-*V* curves of (d) GaAs, (e) PVK and (f) Si with AM 1.5G. The *I*₀-dependent PCE of the (g) GaAs, (h) PVK and (i) Si under lasers with different wavelengths. The lasers are labeled by λ with wavelength marked in subscript. The specific parameters in the figures are attached in the part I of the attachment. The details are shown in supporting information **Supplementary Table 8**.

Table R1. Performance and cost of Photoelectric converters.

Photovoltaic converter	Bandgap adjustment	Flexibility ^a	Heavy Metal ^b	Power per weight (AM 1.5G) ^c (W/g)	Power per weight (Laser) ^d (mW/g)	Cost (CNY/cm ²)
GaAs	Limited	+	○	1.0 ^e	1.3	65.6 ^f
				3.0 ¹	3.9	
Si	Limited	+	×	~0.3 ²	0.4	0.06 ^g
PVK	Limited	++	○	30.3 ³	49.5	2.0 ^h
OLPC	High	+++	×	40.7 ⁴	83.7	4.0 ⁱ

^a: The more the number of “+”, the better the flexibility of photovoltaic converters.

^b: “○” indicates the presence of heavy metal, while “×” indicate the absence of heavy metal.

^c: The power per weight of photoelectric converters reported in the literatures at 100 mW/cm².

^d: The power per weight of photoelectric converters are calculated by the value of literatures, and the illumination intensity of the lasers assumed 10 mW/cm².¹⁻⁴ Due to the lower PCE of commercial cells, the PCEs of GaAs and Si cells are calculated based on the laboratory’s PCEs.

^e: Provided by merchants.

^{f, g}: Provided by merchants, calculated based on the purchase price.

^h: The cost of preparing PVK devices, and the calculation details are shown in supporting information **Supplementary Section S8**.

ⁱ: Literature report.

As shown in **Fig. R1**, in the view point of PCE under low photon flux, GaAs and perovskite LPC (laser power convertor) exhibit better performance than OLPC (organic laser power convertor); while the Si LPC shows even lower PCE than OLPC. Nevertheless, OLPC is still indispensable owing to the absolute advantage in energy budget and laser wavelength adaptability. Based on detail investigation, we summarized the bandgap tunability, flexibility, contain heavy metals or not, power per weight and cost in **Table R1**. Due to the highly adjustable bandgap, ultra-flexible, free of heavy metals, high power per weight, and low cost, OLPCs have unique advantages in wireless power transfer. As shown in **Figure R2**, the large-quantities preparation of OLPC will further reduce the cost of OLPC, which will promote the application of OLPC in wireless power transfer.

To demonstrate the ultra-flexible and lightweight of OPV devices, we prepared the ultra-thin substrate devices based on polyimide (PI). The ultra flexible OPV cells in the literature are also shown in the **Fig. R2**.

Fig. R2. (a) The Schematic diagram of flexible OPV device based on PI. (b) The flexibility display of OPV flexible devices. (c) The ultra-flexible OPV devices reported in the literature.⁵

Fig. R3 Large-quantities preparation of OPV cells. A schematic illustration of knife coating where excess ink is kept ahead of the knife that is in close proximity to the web (left). Slot-die coating relies on the meniscus standing between a coating head with a slot from which ink is supplied to the standing meniscus thus forming a continuous (or striped) wet film (right).⁶

2) On page 1, line 35-36, the authors state that "As photon flux decreases, the PCE of silicon or gallium arsenide photovoltaic cells drops due to the Auger process induced intrinsic losses". The authors should explain why this happens. In principle, the Auger process is a many-body effect that should be more significant under higher photon flux excitation. Additionally, the authors should clarify if there is any particular reason why the Auger process is different in organic devices.

Response: We appreciate the reminding. The statement in the mentioned sentence is replaced in the revised one. We noticed that the intrinsic losses in photovoltaic laser power converters include the entropic loss produced during the radiative recombination and nonradiative recombination.^{7,8} Radiation recombination includes absorption and emission process, while non radiative processes include the free carrier absorption and Auger recombination. In the widely investigated about Si and GaAs, the entropic loss produced during the absorption and emission of radiation is the major loss mechanism for all bandgap energies. In real semiconductor materials, the free carrier absorption and Auger recombination are also unavoidable. Since the free-carrier absorption is a minor effect in Si and GaAs devices, it can be neglected in the calculation.^{9,10} However, Auger recombination can be comparable with radiative recombination even in thin-film materials and thereby is regarded as the sole intrinsic nonradiative recombination in the calculation. It is assumed that the semiconductor material is intrinsic or highly excited. Finite Auger recombination rates will result in a volumetric entropy generation term that determines the entropy generation rate via Auger recombination. Moreover, under higher incident irradiances, the fraction of input power via emission loss is almost constant, while the proportion of entropic loss diminishes logarithmically, as shown in **Fig. R3(a)**.

For diminishing the intrinsic losses with respect to the Auger process: by intensifying the laser irradiance, the proportion of entropic loss in input power can be arbitrarily reduced; by using spectral and angular filters, the intrinsic losses can be diminished via absorption enhancement or emission restriction, as shown in **Fig. R3(b)**. In this case, the excess energy cannot excite new carrier pairs, so that, Auger generation can be neglected. Therefore, the sentence "As photon flux decreases, the PCE of Si and GaAs photovoltaic cells drops due to the Auger process induced intrinsic losses" has been revised to "As photon flux decreases, the PCE of silicon or gallium arsenide photovoltaic cells drops due to the entropic loss induced intrinsic losses". The sentences have been revised and highlighted at line 47-48 in the manuscript.

Fig. R4. Intrinsic losses and transmitted/output power as fractions of input power. (a) The results are given for the Si PLPC with 200 nm thickness illuminated by the 1020 nm laser. (b) The results are given for the GaAs PLPC with 5 nm thickness illuminated by the laser with photon energy equal to the bandgap. For brevity, the entropic loss produced during the absorption and emission of radiation is denoted by “A & E.”⁷

3) On line 107-108, the authors mention that "Only the exciton diffuses to the donor/acceptor interface can be dissociated into free carrier," which is true in conventional devices. However, in OPV devices based on non-fullerene acceptors, charge separation is ready in individual acceptor domains, as evidenced by recent publications such as Nat. Commun. 2022, 13, 2827(2022) or Ref. 26. Therefore, the authors should consider this point.

Response: We appreciate the comment. The regions where exciton dissociates locate at both the bulk and boundary acceptor. As the diffusion length of Frenkel exciton is usually less than tens of nanometers, the exciton dissociation at donor/acceptor interface will take place with lower energy consuming than the one in bulk. Although exciton dissociation can be observed in Y6 based non fullerene receptors (*J. Am. Chem. Soc.* **142**, 12751 (2020).), the efficiency of single component devices is suppressed by bimolecular and rapid minority carrier charge recombination in the absence of donor. Overall, to diminish the ambiguity, we change the sentence to the new one: “**The exciton dissociation into free charge carriers mainly occurs at the donor/acceptor interface.**” at line 109-110 in the manuscript.

4) The authors performed transient absorption measurements with excitation fluence of ~ 10 $\mu\text{J}/\text{cm}^2$, which is far beyond the working condition of the device. In this regime, exciton annihilation cannot be avoided. The authors should rationalize this disparity. Additionally, the authors note that the probe spectrum covers 750-1600 nm, but the data is only plotted in the range < 1100 nm. However, some essential features appear in the longer-wavelength range, and the authors should carefully check for them.

Response: We appreciate the comment. In this work, the radiative intensity of the lasers can reach $300 \text{ mW}/\text{cm}^2$, which equals to $300 \text{ mJ}/(\text{s}\cdot\text{cm}^2)$. So the transient absorption measurements with excitation fluence of $\sim 10 \text{ mJ}/\text{cm}^2$ are comparable to the working condition of the devices.

According to the suggestion, we retested the TA spectrum of PBDB-TF:BTP-eC9 BHJ under excitation of 809 nm, and plotted the spectra at 800-1600 nm in **Fig. R5** (reput in **Fig. 3c** and **Fig. 3e** in the manuscript). As described in the literature,¹¹ the excited-state absorption (ESA) appears at 1400-1600 nm, which is considered to be caused by the intermediate intra-moiety excited (i-EX) state of the hole transfer channel. In addition, the TA spectrum of PB2:GS-ISO BHJ under excitation of 660 nm is shown

in **Fig. R6** (added in supporting information **Supplementary Fig. 7**). However, it is not to appear the feature absorption in the longer-wavelength in PB2:GS-ISO.

Fig. R5. (a) The transient absorption (TA) two dimensional images of PBDB-TF:BTP-eC9 BHJ under excitation of 809 nm, $10.0 \mu\text{J}/\text{cm}^2$. (b) The TA spectra at different delay times of PBDB-TF:BTP-eC9 film with excitation at 809 nm.

Fig. R6. (a) The transient absorption (TA) two dimensional images of PB2:GS-ISO BHJ under excitation of 660 nm, $10.0 \mu\text{J}/\text{cm}^2$. (b) The TA spectra at different delay times of PB2:GS-ISO film with excitation at 660 nm.

Reviewer #2 (Remarks to the Author):

The authors obtained interesting results with organic power converters for low input power conditions at different wavelengths. Unfortunately, I cannot recommend the paper for publication and significant revision would be required. Furthermore, I do not believe the results are noteworthy or significant enough to warrant publication in Nature Communications.

1) The grammar and the style needs significant revisions. For example, the authors use “beamed power converter” => I think they should remove “beamed”. Instead of “beamed”, maybe use “laser” or “optical”, as it is usual used in the field. Grammatically, I think “beamed power converter” does not make sense.

Response: We appreciate the comment. We have revised the language used in our manuscript as your requirement, and the revised part is marked red in the manuscript. The editing certificate is as follows and the original file is attached as Attachment.

Fig. R7. The editing certificate of the manuscript.

In previous literatures, “laser power beaming”, “laser beam”, “laser power converter” and “optical power converter” were used to describe the wireless power transfer.¹²⁻¹⁵ According the suggestion, we have replaced “beamed” with “laser” in this manuscript.

2) The authors used “highly collimated power transmission” => why “highly”, for example further towards the end of the manuscript, the authors proposed “With respect to practical application, the laser can be alternated by cheap but safe monochromatic light-emitting diode (LED).”, so the collimation certainly need not to be “high”.

Response: Due to the 10 W-660-nm laser generator is very expensive, it is not a common-used instrument in our lab. The use of LED spotlight is to imitate the 10 W-660-nm laser and test the performance of laser power converters (OLPC). In practical application of wireless power transfer (WPT), laser generator will be used rather than LED. In addition, the “highly” is removed in corresponding sentence.

3) The authors further claim, “current beamed power converters are incapable of charging micro-power electronics with low photon flux” => I believe this statement to be not true nor substantiated.

Response: We appreciate the comment. This question is the same as comment proposed by **Reviewer 1# question 1)**. Please see the response.

4) They then use the sentence “The principle of laser selecting is established and the up limit of efficiency is prospected.” => I find this sentence has no meaning or its meaning is very unclear.

Response: We appreciate the comment. We change the origin sentences to the new one: “The laser selection principle is established and the upper limit of efficiency is proposed.” at line 17-18 in the manuscript.

5) In addition, the authors mentioned “36.18% efficiency under 660 nm laser with photon flux of 9.46 mW/cm²” => there are too many digits of precision, all throughout the manuscript, including in Table 1, etc. For example, the real efficiency accuracy is probably at best 1 decimal digit of precision. For example, “0.52 W in 2 meters scale” should be replaced with “0.5 W for a distance of 2 m”.

Response: We appreciate the comment. We changed the significant digits after the decimal point to one digit. But for short circuit voltage (V_{oc}), due to it changes little with the illumination intensity or temperature, it still retains three significant digits after the decimal point.

6) The authors claim, “Our work for the first time manifests the superiority of organic photovoltaic cell in constructing organic beamed power converters and provides a solution for the wireless power transfer of micro-power electronics” I find this claim is misleading and unsubstantiated, for example, the literature shows that it is not superior. For example, Komuro et al published “A 43.0% efficient GaInP photonic power converter under high-power 638 nm laser irradiation of 17 W cm⁻²,” or Fafard et al have shown “At 100 mW of input power, an efficiency of $\text{Eff} = 59.7\%$ is obtained ... The output power (P_{mpp}) reaches 242 mW at 500 mW of input power and 415 mW at 1 W”.

Response: We appreciate the comment. The potential application of organic photovoltaic (OPV) cells in laser wireless power transfer has not been explored. For sure, our reporting is the first one in the field of OLPC. In this work, "Our work for the first time manifests the superiority of organic photovoltaic cell in constructing organic beamed power converters and provides a solution for the wireless power transfer of micro-power electronics" refers to our systematic study of the application of OPV cells in laser wireless power transfer and the achievement of a PCE exceeding 36% under micro-power conditions. Moreover, we analyzed the advantages of OLPC in cost and laser wavelength adaptability. Consequently, our work opened up a bright future for OLPC-based WPT. With updates in materials and device, the PCE of OLPC in WPT would emerge as a promising technology.

7) Furthermore, the authors probably have not demonstrated a real “solution”, because the reliability is still ambiguous. Some stability results are shown in the supplementary material, but a real solution would require a solid reliability study. In addition, the device area considered in this study is order of magnitudes much greater than the area used of the typical semiconductor power converters, for example 20 cm² vs < ~0.2cm², therefore the arguments and the comparisons must be adjusted accordingly.

Response: We appreciate the comment. Our work illustrated the potential of organic semiconductor in converting laser into electricity. Focusing on this point, we carried out systematic researches. We think the “real solution” which is expected by the reviewer is complete engineering of OLPC device or even a whole WPT system. This is not the aim of this work, especially for the OPV society which is still undergoing a potential industrialization. What we did is exploit a new application for OPV, and we find that in low-energy laser conversion, the cost of OLPC is the lowest. The solution for the complete technology contains efforts highly beyond the material engineering organized in this work. The finish of encapsulation, optical control, anti-reflection, pin hole, hydrocooling system, and even the integration with energy storage module construct a real solution for OLPC. These items are obviously out of the work we can initialize now because our work is just the first example of OLPC.

According to the suggestion, we compared GaAs and OLPC(PB2:GS-ISO) with an area of 0.2cm^2 and the results are listed here.

Table R2. Output power of GaAs and OLPC(PB2:GS-ISO) at different illumination intensity.*

Illumination Intensity (mW/cm^2)	GaAs ¹⁶ (mW)	OLPC(PB2:GS-ISO) (mW)
10	0.7	0.7
100	8.5	6.4
1000	98	50
10000	1100	/

*: The device area of GaAs and OLPC are 0.2 cm^2 . The GaAs is single junction cells. The GaAs and OLPC are tested at 808 nm and 660 nm, respectively.

As shown in Table R2, GaAs cells are capable of stronger laser illumination intensity, resulting in higher power output at 0.2 cm^2 . Due to the highly adjustable bandgap, ultra-flexible, free of heavy metals, high power per weight, and low cost, OLPCs have unique advantages in wireless power transfer. some micro sensors, detectors and chips only require a few milliwatts or even microwatts power per use, and they maybe do not require long-term power supply, such as anti-theft or anti-counterfeiting of keys, bank cards (or other important card), passive electronic tags (Ink screen does not consume power during daily display) and on-board ETC etc. Except these consumer electronics, the energy consuming of long-term underwater detectors are usually as low as milliwatt-scale.

8) The authors mentioned, “As photon flux decreases, the PCE of silicon or gallium arsenide photovoltaic cells drops due to the Auger process induced intrinsic losses”. I believe this statement is not true nor substantiated: at low photon flux, good state-of-the-art power converters still have good PCE, and Auger process are not expected to be significant at low densities of charged carriers. The output voltage of state-of-the-art semiconductor power converters is only slightly lower at lower at low photon fluxes, as expected from the slightly lower output voltage expected from an ideal diode behavior.

Response: We appreciate the comment. This question is the same as comment proposed by reviewer 1# question 2). Please see that response.

9) The authors mentioned, “it is of great importance to develop the BPC based on new photovoltaic technology with low cost, broad laser wavelength applicability and high PCE under micro-power density.” I find the only aspect of this statement that could be valid in the context of the results presented in the paper is the “low cost” aspect. Although, no data as such to substantiate the relative low cost aspect is presented in the manuscript.

Response: We appreciate the comment. The description of “low cost” is detailed in reviewer 1# question 1). Please see the response.

10) It would probably be interesting if the authors elaborated more on the stability of the results, for example, in Fig. 1, the FF decreases at optical intensities as low as a few tens of milliwatts per cm^2 . Can the authors comment regarding if the devices show degradation at these intensities? Can the I-V or FF curves be retraced multiple times over extended periods of exposure times.

Response: We appreciate the comment. Under low illuminance, the larger energetic disorder causes severe trap-assist recombination, rapidly decreasing FF in organic

photovoltaic cells (OPV).¹⁷ This phenomenon is also common in OPV under indoor lighting test.¹⁸

According to the suggestion, the FF curves be retraced 5 times over extended periods of exposure times, and the results are as follows:

Fig. R8. The FF curves of (a) PB2:GS-ISO and (b) PBDB-TF:BTP-eC9 are retraced 5 times over extended periods of exposure times.

The semiempirical equations for fill factors(FF) of solar cells¹⁹ is shown in **Table 3**, and the curves of FF plotted as a function of optical intensity is shown in **Figure R9**. When the optical intensities decreases, the FF also decreases.

Table R3. Modified semiempirical equations for fill factors of solar cells described by the equivalent circuit shown in **Fig. R9**.*

$FF_j(v_{oc}, \rho_j, a_j, b_j) = \left\{ 1 + a_j \rho_j \exp\left(-\frac{v_{oc}}{b_j}\right) \right\} FF_j^{(0)}(v_{oc}, \rho_j), v_{oc} \geq 1$					
j	Cases	ρ_j	a_j	b_j	$FF_j^{(0)}$
0	$r_s=1/r_p=0$	1	1.07	1.0	$\frac{v_{oc} - \ln(v_{oc} + 0.72)}{v_{oc} + 1}$
S	$0 \leq r_s \leq 0.4, 1/r_p=0$	r_s	1.30	2.0	$FF_0(1-1.1r_s)+0.19r_s^2$
SP	$0 \leq r_s+1/r_p \leq 0.4$	$1/r_p$	0.75	1.5	$FF_s \left\{ 1 - \frac{v_{oc} + 0.7 FF_s}{v_{oc} r_p} \right\}$

*: normalized open-circuit voltage $v_{oc} = \frac{V_{oc}}{nk_B T}$, wherein V_{oc} is the open circuit voltage; n is the diode ideality factor; k_B is the Boltzmann constant; T is the temperature; e is the elementary charge. The r_s and r_p are normalized series and shunt resistances defined by R_s/R_{CH} and R_p/R_{CH} , respectively, in which R_{CH} , the characteristic resistance, is defined as $V_{oc}/(|J_{sc}|A)$.

Fig. R9. Simulated photovoltaic parameters as a function of light intensity I_L :FF. In all cases, the open circles are experimental data, and the lines are the calculated values using Eq. (9), with α, β , and R_s as specified in the bottom part of the figure. Unless specified as zero, $\alpha = J_{SL}(I_L)/I_L = 7.5 \times 10^{-4}$ A/W, $\gamma = [R_{PL}(I_L)A]^{-1}/I_L = 0.030 \Omega^{-1}W^{-1}$, $\beta = J_{ph}(I_L)/I_L = 0.15$ A/W, and $R_s A = 2.2 \Omega cm^2$ were used. J_{s0} , n_0 , and $R_{p0} A$ are $2.3 \mu A/cm^2$, 1.8, and $40, 100 V cm^2$, respectively.¹⁹

11) Why use the scientific notation for the vertical scale of Fig. 2d?

Response: We appreciate the comment. The temperature-dependent EQE values for all the devices can be fitted using the following equation²⁰:

$$\text{EQE} = \text{EQE}_0 \exp\left(-\frac{E_a}{k_B T}\right) \quad \text{equation (R1)}$$

To get E_a , we rewrite the above formula as:

$$\ln \text{EQE} = \ln \text{EQE}_0 - \frac{E_a}{k_B T} \quad \text{equation (R2)}$$

where EQE_0 is the EQE value at infinite temperature, E_a is the activation energy, k_B is the Boltzmann constant, and T is the temperature. The E_a value indicates the energy required for the geminate pair separation. Therefore, the E_a can be fitted with temperature dependent J - V curves.

12) Fig. 2 and 3 show the results of time-resolved and low temperature photocarrier experiments, while the results can be interesting to better understand the carrier dynamics in these materials, they do not demonstrate the “superiority” of the power converters or the efficient power transfer. Fig. 4e shows an I-V curve of a 20cm² module at 2m with an efficiency of 26%, the author should clearly mention the input wavelength for Fig. 4e, as well as the interconnection architecture that leads to a Voc of ~12V. I do not see any superiority of the results presented in Fig 4e compared to other state-of-the-art power converters, other of course than potentially the low cost of the organic materials used.

Response: We appreciate the comment. The results of time-resolved and low temperature photocarrier experiments in this work shown the feasibility of OPV cells application in WPT. What is more important, we explored the exciton annihilation and carrier recombination process of OPV cells under different monochromatic excitation wavelengths.

Regarding the “superiority” of the OLPC, we summarized the bandgap tunability, flexibility, contain heavy metals or not, power per weight and cost of different power converters in **Table R1** (we have elaborated on the advantages of OPV at **reviewer 1# question 1**), please see the response). Due to the highly adjustable bandgap, ultra-flexible, free of heavy metals, high power per weight, and relatively low cost, OLPCs have unique advantages in wireless power transfer.

As shown in **Fig. 1e** and **Fig. 4e** in manuscript, the efficiency of OLPC in WPT can achieve to ~30% PCE at ~100 mW/cm², and the illumination intensity is equivalent to AM 1.5G. Although the PCEs of the PBDB-TF:BTP-eC9-based OLPC have demonstrated the application potential of OPV cells in WPT, there is still much room for PCE improvement. Under the optimal condition of λ_{809} for the PBDB-TF:BTP-eC9-based OLPC, the V_{loss} is as large as 560.0 mV. Moreover, the EQE and FF are still insufficient. Here, the ideal PCE for each laser wavelength is mapped in Fig. 4a and Fig. 4b. By using BHJ materials with lower V_{loss} , higher EQE and larger E_g , the PCE of the OLPC can potentially be significantly improved. loss is as large as 560.0 mV. Moreover, the EQE and FF are still insufficient. Here, the ideal PCE for each laser wavelength is mapped in **Fig. R10** (reput **Fig. R10 (a)** and **Fig. R10 (b)** in **Fig. 4a** and **Fig. 4b** in the manuscript). By using BHJ materials with lower V_{loss} , higher EQE and larger E_g , the PCE of the OLPC can potentially be significantly improved.

Fig. R10. (a) The relationship among PCE, V_{loss} ($V_{\text{loss}}=1240/\lambda-qV_{\text{oc}}$, where q is electron charge) and EQE (a) 70%, (b) 80% and (c) 90%, assuming the E_g of BHJ can be tuned to the value of photon energy of laser. (details of calculation are provided in supporting information **Section S7**). The points for the studied OBPCs are marked in this figure.

According to the suggestion, the schematic diagram is shown in Fig. R11 (reput in supporting information Supplementary Fig. 12), and module fabrication process of 20 cm² module have been added at line 253-259 in the manuscript.

Fig. R11. Schematic diagram of 20 cm² module.

Fabrication of 20 cm² OPV modules. Glass/ITO/PEDOT:PSS/PB2:GS-ISO/PFN-Br/Ag were fabricated by blade coating method. The ITO substrates were purchased from Huananxiangcheng Inc, and patterned with 12 μm P1. ITO substrates were cleaned by above method. The area of mask is 20 cm². After coating and annealing the active layer, the PFN-Br was coated. P2 pattern was formed by a mechanical scribing machine with 50 μm scribing blade. The samples were transferred to thermal evaporator and 150 nm of Ag were deposited. Then, the P3 patterns were formed by a mechanical scribing machine. Notable, to prevent the etched silver from sticking and causing a short circuit, the air knife was applied during the etching process.

Reviewer #3 (Remarks to the Author):

Re: Organic Beamed Power Converter for Efficient Wireless Micro Power Transfer, by Yafei Wang, et al.

Since the first days of organic photovoltaics (OPVs), our community has searched for applications that might demonstrate their unique advantages. Typically, OPV arguments have been based on proposed cost benefits. But over the years, some have also noted that OPVs can outperform conventional cells at low brightness.

1) The physical foundation for interest in performance at low brightness is the localization of optical and electrical excited states in organic materials. This can provide protection from traps, defects and recombination losses. But localization also creates problems at higher optical densities, like lower charge carrier mobilities, and losses such as exciton-exciton and exciton-charge annihilation.

Response: We appreciate the comment. We believe that the illumination intensity at 10^{-1} to 10^2 mW/cm² is the best working condition for OLPC. For sure, localization will affect the performance of OLPC under high illumination intensity. However, the illumination intensity of 10^{-1} to 10^2 mW/cm² has limited impact on performance of OLPC. Within this illumination intensity range, the impact of localization on the performance of OLPC is limited

2) To the best of my knowledge, arguments for the use of OPVs at low brightness never caught on. The question is why would anyone care about a solar cell at low brightness? Maybe there is some benefit in the morning or evening, or maybe the objective is energy harvesting from ambient light? But on the other hand (at least for OPVs), we can't ignore performance at high optical flux either because those are the crucial conditions for actually generating power.

Response: We appreciate the comment. Based on this comment, we have mapped various applications of laser wireless power transfer at different laser wavelengths and output powers in **Fig. R12** (reput in supporting information **Section S1**).

Fig. R12. Illustration of the various WPT applications organized according to the output power (horizontal axis—log scale) and the wavelength of monochromatic (vertical axis—not to scale). The commercial aspects, the reliability, and the technical attributes of the available laser diode products often predominantly guide the selection of the optical input wavelength. We classify the LPC devices into low/regular/medium/high power, based on their output power capabilities.²¹

As shown in **Fig. R12**, some micro sensors, detectors and chips only require a few milliwatts or even microwatts power per use, and they maybe do not require long-term power supply, such as anti-theft or anti-counterfeiting of keys, bank cards (or other important card), passive electronic tags (Ink screen does not consume power during daily display) and on-board ETC etc. Except these consumer electronics, the energy consuming of long-term underwater detectors are usually as low as milliwatt-scale. The charge for these electronics could be convenient when using power transfer. Therefore, the importance of developing BPC suitable for 10^{-1} to 10^2 mW/cm² (Low power) WPT will be unfolded rapidly as the applications of IoTs expand. On the other hand, the use of OPV cells at low brightness, such as the application of OPV cells in indoor LED

light, has attracted the attention of relevant researchers and received systematic research in recent years.^{18,22-25}

In order to obtain higher output power, the laser intensity used can reach tens or even thousands of suns at present. But in our daily life, we need low power intensity to avoid laser damage human bodies and the environment (such as fires, light pollution, etc.). Due to the advantages of light weight, flexibility, low price and high power per weight, OPV cells are more suitable for powering low-power electronics under low light intensity. As shown in **Fig. R10** (at reviewer 2# question 11), please see the response), owing to the adjustable band gap of OPV materials, we firmly believe that the power conversion efficiency of OPV in wireless power transfer system will be further improved with the development of OPV materials.

3) This manuscript presents a variation on this old argument, but applied now to wireless power transfer. The OPVs are indeed outstanding: 17% under AM1.5, and even better under monochromatic excitation closer to the absorption edge. Two main materials are employed, with the wider gap choice unsurprisingly outperforming the conventional OPV structure. The OPV material system is also characterized using TA, which is nice although not especially relevant to the PB2:GS-ISO system that is highlighted as the optimal solution at 660nm.

Response: We appreciate the comment. As shown in **Fig. R12**, with the development of science and technology, the requirements for low power and visible wavelengths of WPT are gradually being proposed. Based on the advantages of OLPC described in **reviewer 1# question 1)**. (please see the response), OLPC is expected to become another highly anticipated devices.

The TA characterization of PB2:GS-ISO have been added in the manuscript **Fig. 4f** to **Fig. 4j**, and the relevant description was added at line 175 to 182. The exciton diffusion length (L_D) in λ_{533} -excited PB2 (**Fig. R14(a)**) and λ_{809} -excited GS-ISO (**Fig. R14(b)**) are 11.8 and 27.6 nm, respectively. The same as BTP-eC9, the longer L_D of GS-ISO accounts for a more efficient exciton diffusion and illustrates the superiority of λ_{660} .

In the existing OPV material system, PB2: GS-ISO is the best choice at 660 nm. However, as shown in **Fig. R12** (at **reviewer 2# question 11**), please see the response), when the EQE and V_{loss} of OPV materials decrease, there will be a higher PCE for OLPC at 660 nm.

Fig. R13 Singlet-singlet exciton annihilation (SSA) decay in neat films of **a**, PB2 (excitation wavelength is 533 nm) and **b**, GS-ISO (excitation wavelength is 660 nm).

Fig. R14. The transient absorption (TA) two dimensional images of PB2:GS-ISO BHJ under excitation of (a). 533 nm, $10.0 \mu\text{J}/\text{cm}^2$ and (d). 660 nm, $10.0 \mu\text{J}/\text{cm}^2$. The TA spectra at different delay times of PB2:GS-ISO film with excitation at (b). 533 nm and (e). 660 nm, respectively. Normalized femtosecond TA exciton dynamics excited at (c). 533 nm with $10.0 \mu\text{J}/\text{cm}^2$ (probed at 630 nm) and (f). 809 nm with $10.0 \mu\text{J}/\text{cm}^2$ (probed at 660 nm) for PB2:GS-ISO film.

4) My main concern with the manuscript is the engineering argument, especially the justification for the key conclusion 'We demonstrate the superiority of constructing OBPC with OPV cell for WPT technology.' I couldn't find any comparisons between the OPV systems and conventional cells, and while the performance sounds great in isolation, one has to wonder what GaAs can do at 660nm. Are we going to exclude the use of Si or GaAs based on cost? I wonder because WPT may not need a large area cell. I also don't understand whether low power, visible spectrum WPT is demanded by any practical application. The last line of the preceding paragraph seems crucial 'The performance of the LED charging system can fulfill the requirement of micro-power electronics such as passive electronic tag, on-board ETC and microfluid chip.' It struck me that a different way to present this work might have been to start with that target application, justify its significance with citations, explain why it needs to be visible spectrum, LED-powered, etc... and then work through the quantitative benefits of this OPV system versus conventional alternatives. I would really like to believe that OPV has a compelling application in WPT, but this manuscript unfortunately does not make that case.

Response: We appreciate the comment. At the response to **Reviewer 1# question 1)**, we compared the performance in low photon flux WPT by using single-junction GaAs, PVK and monocrystalline Si photoelectric converter at λ_{660} and λ_{809} . Based on detail investigation, we summarized the bandgap tunability, flexibility, contain heavy metals or not, power per weight and cost in **Table R1**. Due to the highly adjustable bandgap, ultra-flexible, free of heavy metals, high power per weight, and low cost, OLPCs have unique advantages in wireless power transfer. Please see the response.

Divided by the order of WPT photon flux, the target application scenario varies a lot: 10^3 to $10^5 \text{ mW}/\text{cm}^2$ -WPT techniques often participate in military and spatial applications; 10^2 to $10^3 \text{ mW}/\text{cm}^2$ -WPT techniques are suitable for powering high-energy-consuming loads; as for 10^{-1} to $10^2 \text{ mW}/\text{cm}^2$, few actual products appear in the past years because the low-energy-consuming flexible electronics just enter the

explosive period of developing. Compared with the laser power density in other target application, the low-energy-consuming as for 10^{-1} to 10^2 mW/cm² is “micro power”.

The low power applications for WPT:

1. Power for micro sensors, detectors and chips.
Some micro sensors, detectors and chips only require a few milliwatts or even microwatts power per use, and they maybe do not require long-term power supply, such as anti-theft or anti-counterfeiting of keys, bank cards (or other important card), passive electronic tags (Ink screen does not consume power during daily display) and on-board ETC etc.
2. Power beaming for low power electronics.
The energy consuming of long-term underwater detectors are usually as low as milliwatt-scale. The charge for these electronics could be convenient when using power beaming.

The advantages of WPT within the visible spectrum lasers are:

1. higher PCE:
Similar to the trend of GaAs-series cells that exhibiting higher PCE when replacing IR laser/GaAs cell by green laser/InGaP cell, OPV cells based on BHJs with higher band gaps can output higher PCE if we chose appropriate laser.
2. Higher identification:
When applied to anti-theft or anti-counterfeiting of keys and bank cards (or other important card) etc., it is better to use lasers within the visible spectral in order to visually monitor the process when the holder unlocks or swipes the card. After integration with OLPT, it cannot be used normally even if the thieves copy the key or bank card.
3. Higher safety:
Lasers within the visible spectral range can remind people to avoid the laser.
4. Underwater adaptability:
The penetration of visible spectrum lasers is stronger underwater, especially the blue-green spectrum. The visible spectrum lasers could power for underwater electrical appliances.

According the suggestion, we revised the introduction in this manuscript and marked in highlight at line 33-44, as below:

The target application scenario is divided by the order of WPT photon flux and varies greatly: 10^3 to 10^5 mW/cm²-for WPT techniques that often participate in military and spatial applications; 10^2 to 10^3 mW/cm²-for WPT techniques are suitable for powering high-energy-consuming loads such as cameras or small unmanned aerial vehicles; and 10^{-1} to 10^2 mW/cm²for a few products that have appeared in recent years due to the low-energy-consuming flexible electronics that have entered rapid development period. Some micro sensors, detectors and chips only require a few milliwatts or even microwatts of power per use, and they do not potentially require a long-term power supply, such as anti-theft or anti-counterfeiting keys, bank cards (or other important cards), passive electronic tags (the ink screen does not consume power during daily display) and onboard ETC etc. Except for these consumer electronics, the energy consuming of long-term underwater detectors are usually as low as milliwatt-scale. The charge for these electronics could be convenient when using power beaming. Therefore, the importance of developing LPC suitable for 10^{-1} to 10^2 mW/cm² WPT will unfold rapidly as the applications of IoT expand.

References

- R1 Schoen, J. *et al.* Improvements in ultra-light and flexible epitaxial lift-off GaInP/GaAs/GaInAs solar cells for space applications. *Prog. Photovoltaics* 30, 1003-1011, (2022).
- R2 Rath, J. K., Brinza, M., Liu, Y., Borreman, A. & Schropp, R. E. I. Fabrication of thin film silicon solar cells on plastic substrate by very high frequency PECVD. *Sol. Energy Mater. Sol. Cells* 94, 1534-1541, (2010).
- R3 Wu, J. *et al.* Ultralight flexible perovskite solar cells. *Sci. China Mater.* 65, 2319-2324, (2022).
- R4 Zheng, X. *et al.* Versatile organic photovoltaics with a power density of nearly 40 W g⁻¹. *Energy Environ. Sci.* 16, 2284-2294, (2023).
- R5 Rich, S. I., Lee, S., Fukuda, K. & Someya, T. Developing the Nondevelopable: Creating Curved-Surface Electronics from Nonstretchable Devices. *Adv. Mater.* 34, (2022).
- R6 Sondergaard, R., Hosel, M., Angmo, D., Larsen-Olsen, T. T. & Krebs, F. C. Roll-to-roll fabrication of polymer solar cells. *Mater. Today* 15, 36-49, (2012).
- R7 Lin, M., Sha, W. E. I., Zhong, W. & Xu, D. Intrinsic losses in photovoltaic laser power converters. *Appl. Phys. Lett.* 118, (2021).
- R8 Devos, A. & Pauwels, H. ON THE THERMODYNAMIC LIMIT OF PHOTO-VOLTAIC ENERGY-CONVERSION. *Applied Physics* 25, 119-125, (1981).
- R9 Kosten, E. D., Atwater, J. H., Parsons, J., Polman, A. & Atwater, H. A. Highly efficient GaAs solar cells by limiting light emission angle. *Light-Sci. Appl.* 2, (2013).
- R10 Badescu, V., Landsberg, P. T., De Vos, A. & Desoete, B. Statistical thermodynamic foundation for photovoltaic and photothermal conversion. IV. Solar cells with larger-than-unity quantum efficiency revisited. *J. Appl. Phys.* 89, 2482-2490, (2001).
- R11 Perdigon-Toro, L. *et al.* Barrierless Free Charge Generation in the High-Performance PM6:Y6 Bulk Heterojunction Non-Fullerene Solar Cell. *Adv. Mater.* 32, (2020).
- R12 Shan, T. & Qi, X. Design and optimization of GaAs photovoltaic converter for laser power beaming. *Infrared Phys. Techn.* 71, 144-150, (2015).
- R13 Fafard, S. *et al.* High-photovoltage GaAs vertical epitaxial monolithic heterostructures with 20 thin p/n junctions and a conversion efficiency of 60%. *Appl. Phys. Lett.* 109, (2016).
- R14 Zhou, W. & Jin, K. Optimal Photovoltaic Array Configuration Under Gaussian Laser Beam Condition for Wireless Power Transmission. *IEEE T. Power Electr.* 32, 3662-3672, (2017).
- R15 Schubert, J. *et al.* High-Voltage GaAs Photovoltaic Laser Power Converters. *IEEE T. Electron Dev.* 56, 170-175, (2009).
- R16 Hoehn, O., Walker, A. W., Bett, A. W. & Helmers, H. Optimal laser wavelength for efficient laser power converter operation over temperature. *Appl. Phys. Lett.* 108, (2016).
- R17 Kuik, M., Koster, L. J. A., Wetzelaer, G. A. H. & Blom, P. W. M. Trap-Assisted Recombination in Disordered Organic Semiconductors. *Phys. Rev. Lett.* 107, (2011).
- R18 Wang, W. *et al.* High-performance organic photovoltaic cells under indoor lighting enabled by suppressing energetic disorders. *Joule* 7, (2023).

- R19 Yoo, S., Domercq, B. & Kippelen, B. Intensity-dependent equivalent circuit parameters of organic solar cells based on pentacene and C-60. *J. Appl. Phys.* 97, (2005).
- R20 Gao, F., Tress, W., Wang, J. & Inganas, O. Temperature Dependence of Charge Carrier Generation in Organic Photovoltaics. *Phys. Rev. Lett.* 114, (2015).
- R21 Fafard, S. & Masson, D. P. Perspective on photovoltaic optical power converters. *J. Appl. Phys.* 130, (2021).
- R22 Cui, Y. *et al.* Accurate photovoltaic measurement of organic cells for indoor applications. *Joule* 5, 1016-1023, (2021).
- R23 Wu, Q. *et al.* High-performance organic photovoltaic modules using eco-friendly solvents for various indoor application scenarios. *Joule* 6, 2138-2151, (2022).
- R24 You, Y.-J. *et al.* Highly Efficient Indoor Organic Photovoltaics with Spectrally Matched Fluorinated Phenylene-Alkoxybenzothiadiazole-Based Wide Bandgap Polymers. *Adv. Funct. Mater.* 29, (2019).
- R25 Cui, Y. *et al.* Wide-gap non-fullerene acceptor enabling high-performance organic photovoltaic cells for indoor applications. *Nat. Energy* 4, 768-775, (2019).

REVIEWERS' COMMENTS

Reviewer #1 (Remarks to the Author):

The authors have addressed most of my concerns.

Reviewer #2 (Remarks to the Author):

In this revised version, the authors did good improvements to the manuscript in implementing the reviews' previous comments. I could agree to recommend publication if the authors can implement the following easy but important additional corrections, to further improve the style and readability =>

“However, current laser power converters are incapable of charging micro-power electronics with low photon flux”

Change to:

“However, charging micro-power electronics with low photon flux can be challenging for current laser power converters”

“Here we show the superiority of constructing laser power converters with organic photovoltaic cells for their application in laser wireless power transfer”

Change to:

“Here we show laser power converters with organic photovoltaic cells with good performance for application in laser wireless power transfer”

“The laser selection principle is...” change to

“The laser selection strategy is...”

“a 36.18% efficiency at a 660 nm laser with a photon flux of 9.46 mW/cm²” change, in 2 instances in the text, to

“a 36.2% efficiency at a 660 nm laser with a photon flux of 9.5 mW/cm²” (i.e. adjust proper significant digits)

“For the first time, this work shows the superiority of organic photovoltaic cells in constructing organic laser power converters and provides a solution for the wireless power transfer of micro-power electronics”, change to

“This work shows the good performance of organic photovoltaic cells in constructing organic laser power converters and provides a potential solution for the wireless power transfer of micro-power electronics”

“which is confirmed by the 33.90% PCE” change to “which is confirmed by the 33.9% PCE” (i.e. adjust proper significant digits)

Fig. 1: the legend font size easily can and should be made much larger to be readable

“would induce quasi-single-component excitation” => what is the meaning, I do not understand

The vertical scale of Fig. 2d needs to read “50”, “55”, “60” instead of “ 5×10^1 ”, “ 5.5×10^1 ”, etc

“According to references 38,39” change to “according to results from the literature 38,39”

“The PB2:GS-ISO-based OLPC shows outstanding stability” change to “The PB2:GS-ISO-based OLPC shows good stability”

“We demonstrate the superiority of constructing OLPCs with OPV cells for WPT technology” change to “We demonstrate OLPCs with OPV cells for WPT technology with good performance at low optical intensities”

“Moreover, this work illuminates the superiority of the OPV cells in constructing LPCs for wireless micro-power transmission.” Change to “Moreover, this work illustrates the applicability of the OPV cells as LPCs for wireless micro-power transmission.”

“We establish the laser selection principle based on systematic photophysical studies” change to

“We propose the laser selection strategy based on systematic photophysical studies”

New reference “13. Fafard, S. et al. High-photovoltage GaAs vertical epitaxial monolithic heterostructures with 20 thin p/n junctions and a conversion efficiency of 60%. Appl. Phys. Lett. 109 (2016).” Is a repeat of reference #21 and should instead be replaced with

“13. S. Fafard et al, “Ultrahigh efficiencies in vertical epitaxial heterostructure architectures,” Appl. Phys. Lett. 108, 071101 (2016)”

Reviewer #3 (Remarks to the Author):

The revised version is substantially improved, especially in the SI, where there are now extensive comparisons between the OPVs and alternative technologies. Also, the application is more clearly defined. With these additions, I am confident that the manuscript will provide benefits to the community.

On the other hand, the manuscript still contains some unnecessary hype. For example, the statements in the abstract and conclusion about the 'superiority' of OPVs are not supported by the data. In addition, the key sentence 'the superiority of organic photovoltaic cells in constructing organic laser power converters...' doesn't make sense. How can you make an 'organic laser power converter' without an OPV?

The best summary for the abstract and conclusion might be the one used in the SI 'Due to the [] adjustable bandgap, [flexibility], [absence] of heavy metals, high power per weight, and low cost, OLPCs have unique advantages in wireless power transfer'. [Note that I edited the original sentence.]

The key point is that the OPVs have unique and maybe useful properties. This manuscript at its best summarizes those properties and compares them to alternative technologies.

Finally, a couple of minor comments.

Supplementary Fig. 1 could be a useful cartoon for summarizing applications. But I don't understand the wavelength dependence of the blobs. Why are some applications limited to a long-wavelength cutoff of ~ 900nm? Why are others limited to a fairly narrow range of 950-800nm (is that because they must be non-visible and silicon compatible?). The caption says 'The commercial

aspects, the reliability, and the technical attributes of the available laser diode products often predominantly guide

the selection of the optical input wavelength. ' Not sure that will explain the issue sufficiently. If there isn't room to explain perhaps a relevant citation could be provided?

Some of the legends were illegible in Supplementary Figs 16 and 17. I wrote this review assuming that the wavelengths were 809nm (blue) and 660nm (orange).

Response to decision letter (purple for comments; black for responses; red for revisions).

Reviewer #1 (Remarks to the Author):

The authors have addressed most of my concerns.

Response: We would like to sincerely thank the reviewers for spending precious time on this paper and providing invaluable comments which substantially helped improving the quality of the paper.

Reviewer #2 (Remarks to the Author):

In this revised version, the authors did good improvements to the manuscript in implementing the reviews' previous comments. I could agree to recommend publication if the authors can implement the following easy but important additional corrections, to further improve the style and readability =>

1. "However, current laser power converters are incapable of charging micro-power electronics with low photon flux" change to:

"However, charging micro-power electronics with low photon flux can be challenging for current laser power converters".

Response: We appreciate the recommendation. The sentence has been changed and highlighted at line 16-17 in the manuscript.

2. "Here we show the superiority of constructing laser power converters with organic photovoltaic cells for their application in laser wireless power transfer" change to:

"Here we show laser power converters with organic photovoltaic cells with good performance for application in laser wireless power transfer".

Response: We appreciate the recommendation. The sentence has been changed and highlighted at line 17-18 in the manuscript.

3. "The laser selection principle is..." change to:

"The laser selection strategy is...".

Response: We appreciate the recommendation. The sentence has been changed and highlighted at line 18-19 in the manuscript.

4. "a 36.18% efficiency at a 660 nm laser with a photon flux of 9.46 mW/cm²" change, in 2 instances in the text, to:

"a 36.2% efficiency at a 660 nm laser with a photon flux of 9.5 mW/cm²" (i.e. adjust proper significant digits).

Response: We appreciate the recommendation. The significant digits of this manuscript have been revised and highlighted in the manuscript.

5. "For the first time, this work shows the superiority of organic photovoltaic cells in constructing organic laser power converters and provides a solution for the wireless power transfer of micro-power electronics", change to:

"This work shows the good performance of organic photovoltaic cells in constructing organic laser power converters and provides a potential solution for the wireless power transfer of micro-power electronics".

Response: We appreciate the recommendation. The sentence have been changed and highlighted at line 21-23 in the manuscript.

6. "which is confirmed by the 33.90% PCE" change to:

“which is confirmed by the 33.9% PCE” (i.e. adjust proper significant digits). We appreciate the recommendation. The significant digits of this manuscript have been revised and highlighted in the manuscript.

7. Fig. 1: the legend font size easily can and should be made much larger to be readable.

Response: We appreciate the recommendation and the legend font size in **Fig. 1** has been adjusted.

8. “would induce quasi-single-component excitation” => what is the meaning, I do not understand.

Response: We appreciate the comment. As shown in **Fig. R1** (**Fig 1b** in manuscript), either PBDB-TF or BTP-eC9 exhibits absorption at 809 nm and 533 nm, respectively. Thus it is challenging to absolutely excite each neat component in PBDB-TF:BTP-eC9 BHJ. What we can do is to select laser wavelength where the absorption of PBDB-TF and BTP-eC9 show substantial difference. The “quasi-single-component excitation” is used to describe the concerning. As ambiguity exists here, we change the sentence to: “**would primarily excite PBDB-TF or BTP-eC9 in BHJ**” and highlighted it at line 95 in revised manuscript.

Fig. R1. Upper item displays the absorption coefficient of neat PBDB-TF, BTP-eC9 and PBDB-TF:BTP-eC9 BHJ, respectively. The bottom item displays the EQE spectra of the OPV under AM 1.5G illumination at 100 mW/m^2 .

9. The vertical scale of Fig. 2d needs to read “50”, “55”, “60” instead of “ 5×10^1 ”, “ 5.5×10^1 ”, etc.

Response: We appreciate the recommendation. The vertical scale of Fig. 2d has been revised to “50”, “55”, “60”, etc.

10. “According to references 38,39” change to: “according to results from the literature 38,39”

Response: We appreciate the recommendation. The sentence has been changed and highlighted at line 154 in the manuscript.

11. “The PB2:GS-ISO-based OLPC shows outstanding stability” change to: “The PB2:GS-ISO-based OLPC shows good stability”

Response: We appreciate the recommendation. The sentence has been changed and highlighted at line 213 in the manuscript.

12. “We demonstrate the superiority of constructing OLPCs with OPV cells for WPT technology” change to:

“We demonstrate OLPCs with OPV cells for WPT technology with good performance at low optical intensities”

Response: We appreciate the recommendation. The sentence has been changed and highlighted at line 227 in the manuscript.

13. “Moreover, this work illuminates the superiority of the OPV cells in constructing LPCs for wireless micro-power transmission.” change to:

“Moreover, this work illustrates the applicability of the OPV cells as LPCs for wireless micro-power transmission.”

Response: We appreciate the recommendation. The sentence has been changed and highlighted at line 228 in the manuscript.

14. “We establish the laser selection principle based on systematic photophysical studies” change to:

“We propose the laser selection strategy based on systematic photophysical studies”.

Response: We appreciate the recommendation. The sentence has been changed and highlighted at line 229-230 in the manuscript.

15. New reference “13. Fafard, S. et al. High-photovoltage GaAs vertical epitaxial monolithic heterostructures with 20 thin p/n junctions and a conversion efficiency of 60%. Appl. Phys. Lett. 109 (2016).” Is a repeat of reference #21 and should instead be replaced with

“13. S. Fafard et al, “Ultrahigh efficiencies in vertical epitaxial heterostructure architectures,” Appl. Phys. Lett. 108, 071101 (2016)”

Response: We appreciate the recommendation. The reference 13 has been replaced and highlighted at line 340-341 in the manuscript.

We would like to sincerely thank the reviewers for spending precious time on this paper and providing invaluable comments which substantially helped improving the quality of the paper.

Reviewer #3 (Remarks to the Author):

The revised version is substantially improved, especially in the SI, where there are now extensive comparisons between the OPVs and alternative technologies. Also, the application is more clearly defined. With these additions, I am confident that the manuscript will provide benefits to the community.

1. On the other hand, the manuscript still contains some unnecessary hype. For example, the statements in the abstract and conclusion about the 'superiority' of OPVs are not supported by the data. In addition, the key sentence 'the superiority of organic photovoltaic cells in constructing organic laser power converters...' doesn't make sense. How can you make an 'organic laser power converter' without an OPV?

Response: We appreciate the comment. We have polished the language in manuscript following the guidance formatting articles. In addition, reviewer 2# gave advice in last review that the key sentence in abstract should be changed to “**This work shows the good performance of organic photovoltaic cells in constructing organic laser power converters and provides a potential solution for the wireless power transfer of micro-**

power electronics”. We followed this suggestion and highlighted it at line 21-23 in the manuscript if reasonable.

2. The best summary for the abstract and conclusion might be the one used in the SI 'Due to the [adjustable bandgap], [flexibility], [absence] of heavy metals, high power per weight, and low cost, OLPCs have unique advantages in wireless power transfer'. [Note that I edited the original sentence.]

The key point is that the OPVs have unique and maybe useful properties. This manuscript at its best summarizes those properties and compares them to alternative technologies.

Response: We appreciate the comment. According to the suggestion, the summary for the abstract and conclusion have been revised and highlighted at line 21-23 and 237-238 in the manuscript.

3. Finally, a couple of minor comments.

Supplementary Fig. 1 could be a useful cartoon for summarizing applications. But I don't understand the wavelength dependence of the blobs. Why are some applications limited to a long-wavelength cutoff of ~ 900nm? Why are others limited to a fairly narrow range of 950-800nm (is that because they must be non-visible and silicon compatible?). The caption says 'The commercial aspects, the reliability, and the technical attributes of the available laser diode products often predominantly guide the selection of the optical input wavelength.' Not sure that will explain the issue sufficiently. If there isn't room to explain perhaps a relevant citation could be provided?

Response: We appreciate the comment. Supplementary Fig. 1 is made according to the reference: Fafard, S. & Masson, D. P. Perspective on photovoltaic optical power converters. *J. Appl. Phys.* **130**, 160901, (2021). The drawing of spot boundaries rigidly follows this reference and acts as a general classification based on the summary of the developing in lasers and laser power converter (LPC) applications to the state of the art. The selection of laser should consider the overall aspects including generator cost, technology maturity, LPC absorption, target application and device performance¹⁻². In terms of LPC absorption and device performance, for example, LPCs show maximum PCEs in the case of the energy of photon is close to the bandgap of laser-absorbing material.³⁻⁴ Due to the different gain media of lasers with different wavelengths, the commercial aspects, the reliability, and the technical attributes of the available laser diode products are different. In the classifications sketched in **Supplementary Fig. 1**, high-power semiconductor lasers with an emission wavelength of 800–980 nm are widely used in the pump systems of solid-state lasers and fiber amplifiers.⁵⁻⁶ It is also compatible with the bandgap of gallium arsenide and silicon cells. For optical power transfer via optical fibers, 1064 nm, 1300 nm and 1550 nm wavelengths are most suitable as radiation at these wavelengths suffers the least attenuation while propagating inside the fiber, thereby increasing the overall system efficiency.⁷⁻⁸ As mentioned above, the selection of laser and the targeted matching towards LPC are dynamical and flexible those depend on available industries and technologies.

4. Some of the legends were illegible in Supplementary Figs 16 and 17. I wrote this review assuming that the wavelengths were 809nm (blue) and 660nm (orange).

Response: We appreciate the comment. The legends in **Supplementary Fig 16** and **17** have been enlarged.

We would like to sincerely thank the reviewers for spending precious time on this paper and providing invaluable comments which substantially helped improving the quality of the paper.

References

1. Fafard, S. & Masson, D. P. Perspective on photovoltaic optical power converters. *J. Appl. Phys.* **130**, 160901, (2021).
2. Fafard, S. et al. Power and Spectral Range Characteristics for Optical Power Converters. *Energies* **14**, 4395, (2021).
3. Jarvis, S. D., Mukherjee, J., Perren, M. & Sweeney, S. J. Development and characterisation of laser power converters for optical power transfer applications. *Iet Optoelectron.* **8**, 64-70, (2014).
4. Lin, M., Sha, W. E. I. Zhong, W., and Xu, D. Intrinsic losses in photovoltaic laser power converters. *Appl. Phys. Lett.* **118**, 104103 (2021).
5. Bett, A. W. et al. in *33rd IEEE Photovoltaic Specialists Conference*. 362-366 (2008).
6. Khvostikov, V. P., Sorokina, S. V., Potapovich, N. S., Khvostikova, O. A. & Timoshina, N. K. Laser ($\lambda=809$ nm) power converter based on GaAs. *Semiconductors* **51**, 645-648, (2017).
7. Khvostikov, V.P., Kalyuzhnyy, N.A., Mintairov, S.A. et al. Module of Laser-Radiation ($\lambda = 1064$ nm) Photovoltaic Converters. *Semiconductors* **53**, 1110–1113 (2019).
8. Mukherjee, J., Jarvis, S., Perren, M. & Sweeney, S. J. Efficiency limits of laser power converters for optical power transfer applications. *J. Phys. D: Appl. Phys.* **46**, (2013).